# Metamodelling for Design of Mechatronic and Cyber-Physical Systems

**Krzysztof Pietrusewicz** 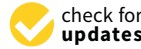

West Pomeranian University of Technology, Szczecin, Sikorskiego 37, 70-313 Szczecin, Poland; krzysztof.pietrusewicz@zut.edu.pl; Tel.: +48-663-398-396

**Featured Application: The presented results were used to design innovative mechatronic solutions: motion control algorithm of 5-axis feed drive "X-5" processing centre owned by the AVIA company, loading and forestry crane control system designed within the EU 7FP IAPP project called iLOAD.**

**Abstract:** The paper presents the issue of metamodeling of *Domain-Specific Languages* (DSL) for the purpose of designing complex mechatronics systems. Usually, one of the problems during the development of such projects is an interdisciplinary character of the team that is involved in this endeavour. The success of a complex machine project (e.g., Computer Numerically Controlled machine (CNC), loading crane, forestry crane) often depends on a proper communication between team members. The domain-specific modelling languages developed using one of the two approaches discussed in the work, lead to a machine design that can be carried out much more efficiently than with conventional approaches. Within the paper, the *Meta-Object Facility* (*MOF*) approach to metamodeling is presented; it is much more prevalent in modern modelling software tools than *Graph-Object-Property-Relationship-Role (GOPRR)*. The main outcome of this work is the first presentation of *researchML* modelling language that is the result of more than twenty ambitious research and development projects. It is effectively used within new enterprises and leads to improved traceability of the project goals. It enables for fully-featured automatic code generation which is one of the main pillars of the agile management within mechatronic system design projects.

**Keywords:** metamodeling; *Meta-Object Facility*; V-model; *INCOSE* process; *MagicGrid* process; *Model-Based Design*; agile management; mechatronic system design; cyber-physical systems; *SCRUM*; *Project Cycle Management*

## 1. Introduction

Development of modern mechatronic systems is often attributed to the emergence of open control systems for CNC machines [1–3]. The complexity of hardware and software architectures used in real-time control systems forced many engineering fields to follow a design process that is organized, systematic and well-structured. Consequently, these paradigms greatly influenced the development of novel mechatronic systems [4–6].

Innovations in new products are foremost motivated by three principles: (1) increase in production efficiency [7]; (2) learning curve reduction, that is, faster product adaptation by the end-user; (3) ensuring safe operation or improvement of the overall safety conditions [8–10]. With the advent of new technologies and the ever-increasing product complexity, the classical document-driven approach towards governing the development process is no longer adequate. Even the nature of development teams has changed thoroughly. They are formed by specialists with diversified areas of expertise who are often spread around the world and work in small local teams; particular tasks are also outsourced



to third party companies. In order to aid themselves in this multifaceted task, the engineers are increasingly exploiting modelling tools and languages.

In the presented context, model is understood as an abstract representation of a system used in object control. Sometimes, these models also include the environment—the surroundings of the modelled system. Closely related to the concept of model is the so called *metamodel* or *model of a model*. It serves as a notation or language describing the model using limited semantics. Metamodels can be either graphical or textual [11] and have been present in the engineering field for many years. But up until recently, they were only used for description and better understanding of various phenomena present in the environment. Today, their development is an essential step during the control system design process [2].

Investment in the model-based development is justified in the following cases: the modelled system is large, complex or cross-domain in nature (e.g., contains electric, hydraulic, pneumatic components), the modelled system is miniaturized, the proof-of-concept experiments are either too time-consuming or expensive and finally, if the developed system is to be modified in the future or is simply unavailable or does not yet exist at the time of control system development.

In References [2,11], the importance of control system design based on models is emphasized (i.e. the controlled object's model and the model of algorithm). However, the use of system modelling must be rational as it consumes both human and material resources and requires the use/presence of sophisticated hardware, software, knowledge, skills or technical prowess of the team. Additionally, systems that are released in one variant and are not a subject to further modifications render the model-based approach ineffective and unnecessary as the potential gain does not outweigh the investment cost.

Section 1 of the following work is concerned with the rationale for model-based approach in development of mechatronic systems. Modelling of mechatronic systems compliant with the *Model-Based Design* paradigm is presented in Section 2. This section also contains information about commonly used tools, approaches and different stages of team's immersion into the adopted paradigm. Section 3 presents the metamodeling aspects and two most popular meta-metamodels: *GOPRR* and *MOF*. In Section 4 two mechatronic systems projects are presented and discussed in Section 5 accordingly. Original generic workflow for *Model-Based Design* is proposed in Section 5. Section 6 presents *researchML* metamodel developed to support mechatronic and cyber-physical system design process. Finally, the paper is concluded in Section 7.

## 2. Modelling Mechatronic Systems

Development of machine control systems is one of the specific areas for which the creation of a dedicated domain modelling language is sensible [12]. In this paper, a *Model-Based Design* (MBD) approach is proposed as a paradigm for designing new control systems [7]. Currently, an effective development of control systems combines modelling tools, computer simulation and automated generation of the control system code. The following issues can be attributed as belonging to the system modelling field [5,13]:

- Requirements modelling—requirements set for the system; a fusion of the adopted business model, law regulations and identified functionalities that are necessary to meet the project's objectives and business goals [4,14,15];
- Interface modelling—information exchange between crucial components of the system and between the system and external environment (including other systems) [16];
- Hardware and software modelling—models of system's software and hardware components; limitations must be considered that stem from the following sources: adopted business model, functional requirements, law regulations;
- Use case modelling—definition of interactions between various client types and the system itself, broad spectrum of use case specificity;

- Control target modelling—modelling of architecture and properties of the controlled object; enables parametrization of the proposed solution in simulations;
- Control software modelling—modelling of architecture and features of the software solution used to control the developed system.

*Model-Based System Engineering* (MBSE) [5,17–19] is a complex modelling process for integration of various mechatronic systems and components. It considers the properties of the control system as well as the workspace specifics in which the combined system will operate. MBSE expands upon the ideas of *Model-Based Design* augmenting it with mathematical formalism [17].

### 2.1. Model-Based Design Paradigm

In Reference [7], the nine-box model of immersion in the *Model-Based Design* process of control algorithm development is presented (Figure 1). The companies with the highest degree of immersion (Figure 1 MBD-9) employ a fully model-based approach during the research and development phase. The bare minimum, however, is to use a graphical representation of the developed control system components (Figure 1 MBD-1).

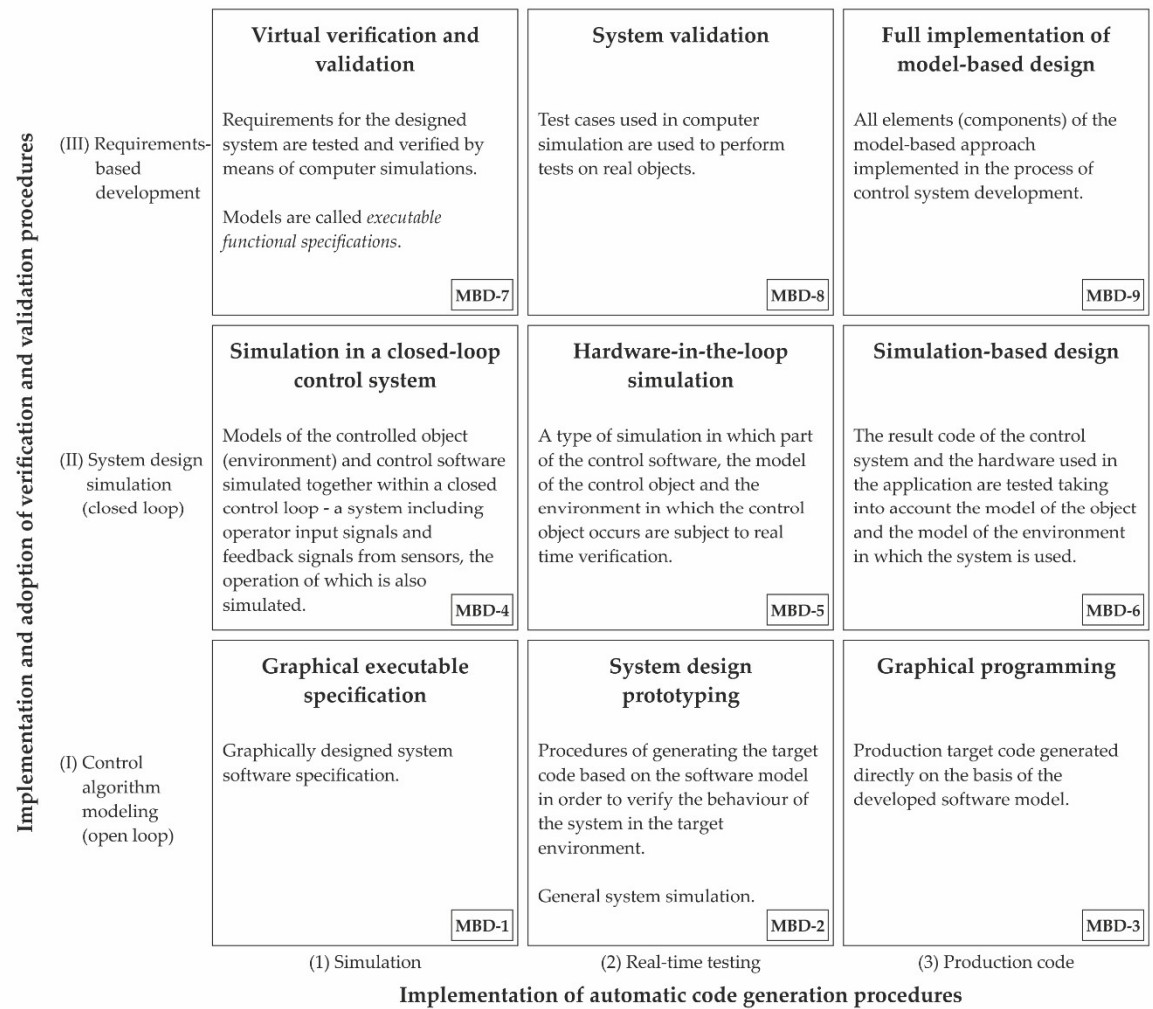

**Figure 1.** The nine-box model for implementation of *Model-Based Design* concepts [7].

The following benefits can be attributed to the MBD approach:

1. Final product's conformity to the specified requirements,
2. Reduction of software errors,

3.  Verification of the project's goals without building a physical prototype,
4.  Development of test-case scenarios and their validation in a simulation environment,
5.  Easier management of complex projects.

Two important areas of mechatronic and cyber-physical systems design are represented in Figure 1 by columns and rows. These are respectively:

1.  Implementation and adoption of verification and validation procedures:

    - (I) Control algorithm modelling (open loop)—no validation support; specifications include only operational level information;
    - (II) System design simulation (closed loop)—closed loop simulations are used to provide feedback data for designers; real-time testing before moving the control system to a target platform allows to eliminate most errors from the final product without the risk of damaging its physical components (e.g., sensors, actuators);
    - (III) Requirements-based development—requirements management is ingrained in the system design process;

2.  Implementation of automatic code generation procedures:

    - (1) Simulation—code generation supports only system and dynamic simulation calculations; no real-time code is generated;
    - (2) Real-time testing—real-time tests are additionally supported by automatic code generation procedures;
    - (3) Production code—this highest level of software development is achieved by utilizing toolchains that support code generation for the final product directly out of the system's specifications.

**Implementation and adoption of verification and validation procedures** (Figure 1) results in a better integration of business goals within system design procedures.

**Implementation of automatic code generation procedures** (Figure 1) improves the process of software development; the development time is reduced and less errors are present in the final product's source code.

Fully adopting the *Model-Based Design* principles is a time-consuming process. Migrating from the process loosely based on models, most likely only used for documenting and organizing the software aspect of the solution (Figure 1. MBD-1), towards one that is interlaced with them on all levels (e.g., law legislation, software and hardware aspects of the solution, business requirements) demands vast investment in both human and material resources.

For most of the companies, adopting the MBD-4 level results in a significant increase in efficiency. It also reduces the risks present during new product development. Higher levels of adoption are a proof that the company is aware of intricacies present in research and development processes and chooses a systemic approach to pursue them.

At the highest level of immersion (represented in Figure 1 by states from MBD-7 to MBD-9), the ability to intertwine the requirements set for a product with the ability to validate and verify them in simulations closely resembling the end-product's working conditions is an unparalleled advantage. This results in a faster development of new product versions, adding features to the existing ones is easier, finally, the adaptation to the ever-changing market conditions (e.g., new legislation) is more straightforward. Therefore, employing such measures gives a cutting edge over competition.

In the following subsections all levels of the *Model-Based Design* adoption presented in Figure 1 are described.

### 2.1.1. Graphical Executable Specification (MBD-1)

If a system is designed with the use of a chosen modelling language (general purpose: *UML*, *SysML*; domain-specific: *EAST-ADL* or any other domain language), then the so-called graphical specifications are created. They are later manually converted into models native to other software applications (e.g., *AutoCAD*, *Simulink*) or produce semantic artefacts (source code, textual and graphical documentation, control system schemas).

### 2.1.2. System Design Prototyping (MBD-2)

The adoption of MBD on this level means that fragments of the graphical specification (MBD-1) are converted into parts of the implementation by the means of code generation. The code generation is tied closely to the notion of an **executable specification**; it allows the verification of the requirements and system's behaviour in the targeted environment. Only then the code for the control system is generated.

The programmer is still responsible for managing the developed software—the final source code is put together manually. However, the entire process involves incorporating large code fragments already supplied by the code generation feature.

### 2.1.3. Graphical Programming (MBD-3)

At this level the code for the control system is generated entirely by the means of graphical specifications. The binary files (as well as other semantic artefacts) are a result of a fully-automated software development process that takes into account the type of target system and its working environment.

### 2.1.4. Simulation in a Closed-loop Control System (MBD-4)

Computer simulations are a great tool to analyse the important aspects of complex systems' control. They also enable verification of the completeness of a software design which is used to generate the control system's source code.

Most often, the simulations include the models of objects and environment and designed control algorithm and are typically deployed on a desktop class PC. Even if a decision is made to separate these models into different software solutions interfaced with one another, they are still simulated in the same hardware environment (i.e. on the same physical device).

Very interesting from an academic point of view is the notion of automatically generated simulation models [20] which are derived from the control system's specification. The integration of MetaEdit+ and Matlab/Simulink software is one such example (http://metacase.com/cases/simulink-integration.html). It is very useful when designing control systems adhering to the ISO 26262 standard [21].

### 2.1.5. Hardware-in-the-loop Simulation (MBD-5)

*Hardware-in-the-loop* (HIL) simulation [22] is a procedure in which a simulation model of a closed control loop is subjected to an automated code generation and later implemented on a hardware working in the real-time regime.

Different to the previous approach (MBD-4), the environment and/or the controlled object are placed on separate hardware entities. Nowadays, this could be one of the fast prototyping platforms (e.g., dSpace, Opal-RT, xPC Target) or one of the Programmable Logic Controller or Programmable Automation Controller (PLC/PAC) [23]. Although the control code is executed on a physical device, there is no danger of damaging the physical components of the system as all the output and input signals are simulated.

### 2.1.6. Simulation-based Design (MBD-6)

The specification of a control system which includes the target's environmental parameters is at first migrated to a simulation environment. Next, the control system's source code is generated along with object's and environment's models. Then, the complete code is tested in Hardware-In-the-Loop (HIL) simulation on a target platform. After successfully passing the tests in HIL simulation, the final source code is generated for the target control system. The only manual activities present at this level are hardware configuration and software configuration of the target platform.

The target platform and its algorithms are validated with two types of tests—simulation tests and computational feasibility in real-time regime. These tests are essential for ensuring the quality of the solution and its infallibility.

### 2.1.7. Virtual Verification and Validation (MBD-7)

This level is the first one where a systemic approach to the requirements management and tests design is taken. The main difference between this level (MBD-7) and the one directly below it (MBD-4) is that the project's requirements are factored into the design process. The simulation models of a closed control loop are based upon them. Minor modifications of these requirements may be necessary after each simulation iteration. This does not, however, change the fact that a successful verification and validation must end with a documentation that will be useful to the designer at later stages.

The test case scenarios and target platform description may differ, that is, the tests are not constrained by the proposed hardware solution. As counterintuitive as it seems, this has one major advantage—extensive simulation tests allow to look into multiple scenarios that would normally be unavailable to verify on a real-world object. Also, the test cases of these two areas are conducted independently with the use of different tools.

### 2.1.8. System Validation (MBD-8)

This level of the MBD adoption is achieved when the test cases for a control's system closed loop are first conducted on a simulation model and later are verified in HIL simulation. Finally, the tests take place on a real-world system. The software implementation of the control system is tested and documented identically as in the case of virtual validation and verification.

### 2.1.9. Full Implementation of *Model-Based Design* (MBD-9)

The last, ninth level of immersion integrates all of the beforementioned stages into a fully-automated design process.

### 2.2. Modelling Workflow

A potential toolchain representing the full implementation of *Model-Based Design* is shown in Figure 2.

Most of the research focus is devoted to the transitions marked as (1), (2) and (3) in Figure 2. It can be explained by the fact that the highest increase in productivity and the end-product's quality is achieved by automation of the following areas of control system design: generation of simulation models, control system source code generation, generation of documentation validating the conformity with the requirements.

Depending on the adoption level of *Model-Based Design*, different tools and approaches (e.g., manual vs. automated) will be utilized to aid the design process found in Figure 2.

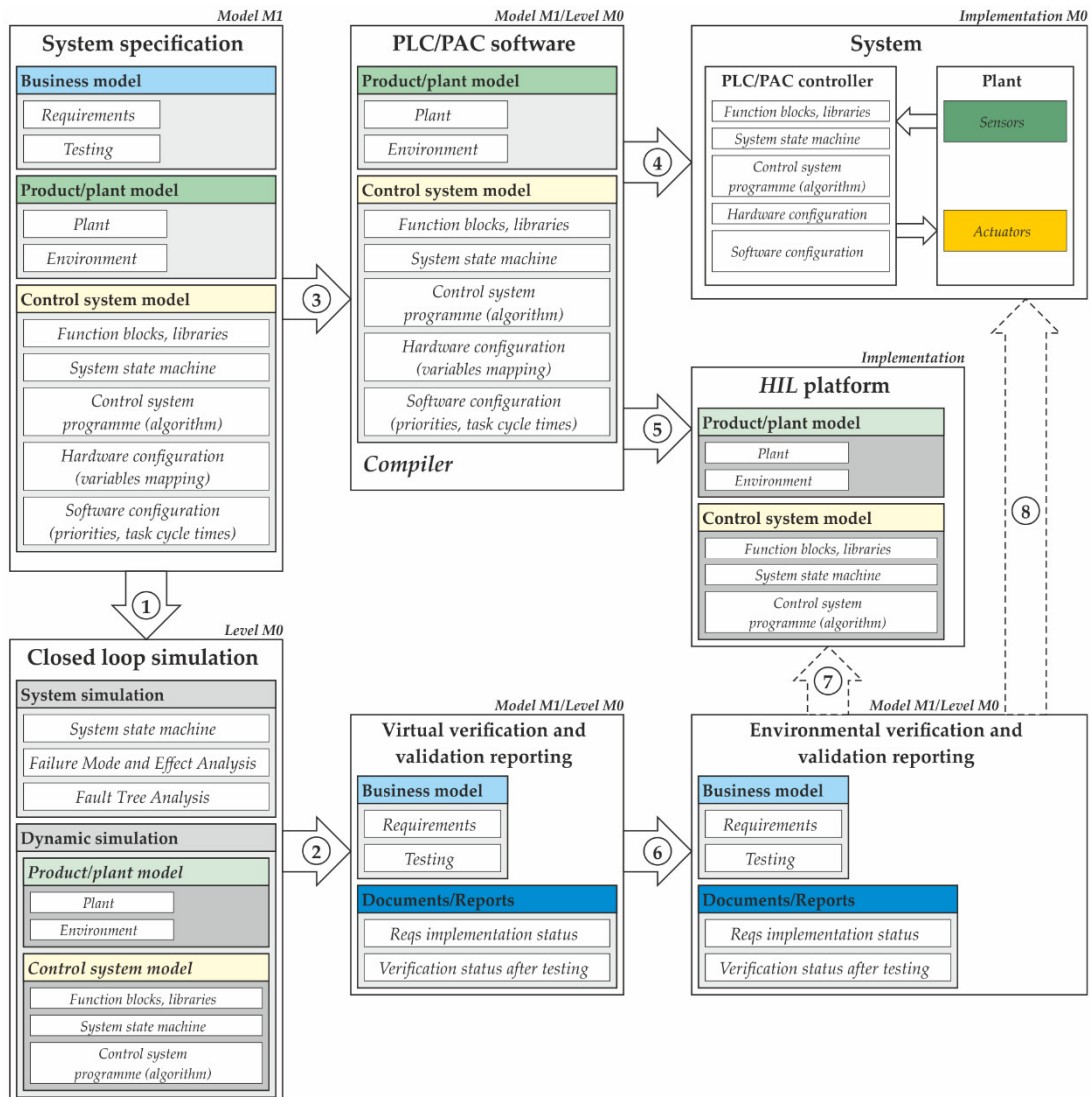

**Figure 2.** The fully-featured (ideal) *Model-Based Design* framework/toolchain/workflow. (1) automatic generation of simulation files [20,24,25]; (2) generating documentation from simulation tests; (3) automatic code generation for PLC/PAC [23,26,27]; (4) programming the control platform; (5) *Hardware-In-the-Loop* testing [28–30]; (6) virtual testing used to validate of the solution [7,18]; (7,8) conducting real-time testing for final system approval.

## 2.3. Modelling Approaches

*MBSE* cannot exist without a proper work methodology. Without a workflow definition the *MBSE* is nothing but a somewhat useful set of tools supporting selected engineering aspects. INCOSE OOSEM and *MagicGrid* from NoMagic are the most popular, well-defined and proven with many industrial case studies.

However, new methodologies appear, as the two methods mentioned above are not providing the best fir for certain niches. One such example is given in Reference [18], where a methodology called *Property-Model Methodology* (PMM) is introduced; it is a safety-oriented combination of property-based requirements and *MBSE*.

### 2.3.1. INCOSE Object-Oriented Systems Engineering Method (OOSEM)

INCOSE OOSEM is based on the functional decomposition approach. It employs the top-down approach, where the modelling order takes place from the most general to the most detailed areas of

interest. *SysML* language is the preferred way for modelling, managing, tracing and documenting projects compliant with the INCOSE methodology.

INCOSE workflow is quite linear and combines requirements model, use cases definition, architecture design and implementation within a single project toolchain as shown in Figure 3. INCOSE methodology workflow is based on a hierarchical realization of the following steps: from modelling and project setup, through analysis, definition of requirements and logical architecture, up to final verification and validation.

INCOSE OOSEM supports identifying risks and potential problems at the early project stages. Steps 1–5 (Figure 3) define the most important modelling stages necessary to fully describe the project and architecture of the implemented solution. These steps are usually the most time and resource consuming among the entire project. Steps 6–10 (Figure 3) are practices and activities conducted during modelling and system development.

INCOSE OOSEM was developed as a systematic alternative to the so-called document-centric approach to project development. INCOSE is used in many Systems Engineering problems [19,31,32]. Unfortunately, it is hard to understand by inexperienced engineers.

### 2.3.2. MagicGrid

In the case of complex mechatronic systems spanning across multiple domains, a much more approachable workflow was proposed by a group of engineers from NoMagic company. *MagicGrid* workflow is shown in Figure 4.

Parametrization and measurement of effectiveness (C4, P4, Figure 4) is the main difference between research done on prototypes (phases 1–10, Figure 4) and development of final products (transfer of scientific results to a business reality, that is, market).

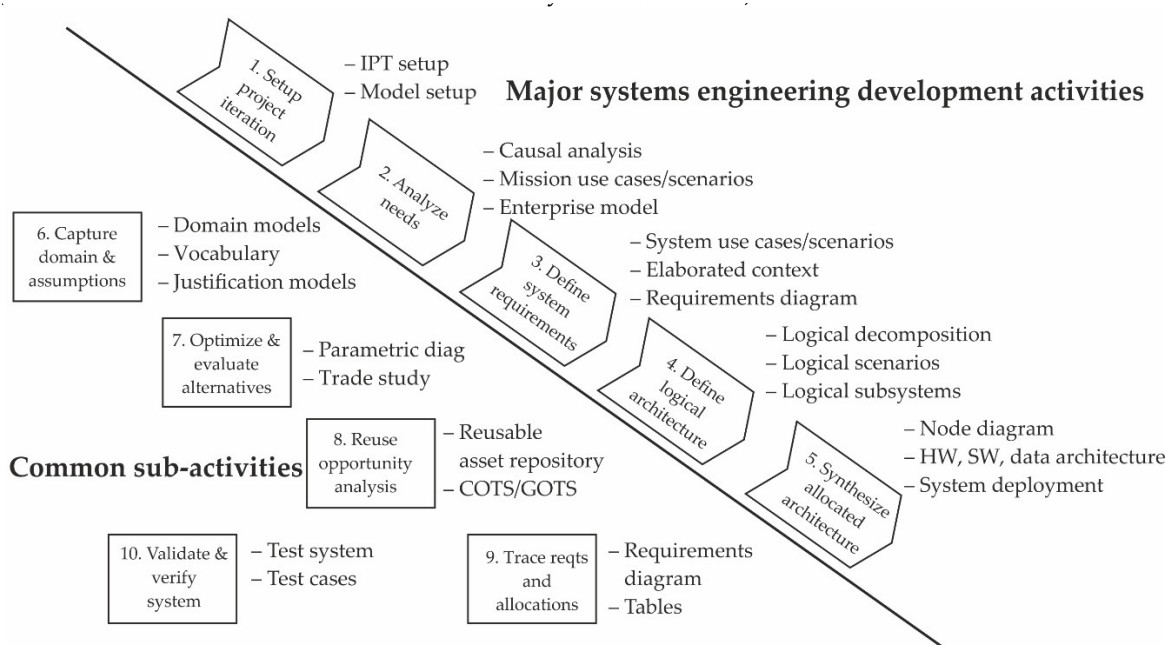

**Figure 3.** INCOSE OOSEM in Model-Based Systems Engineering.

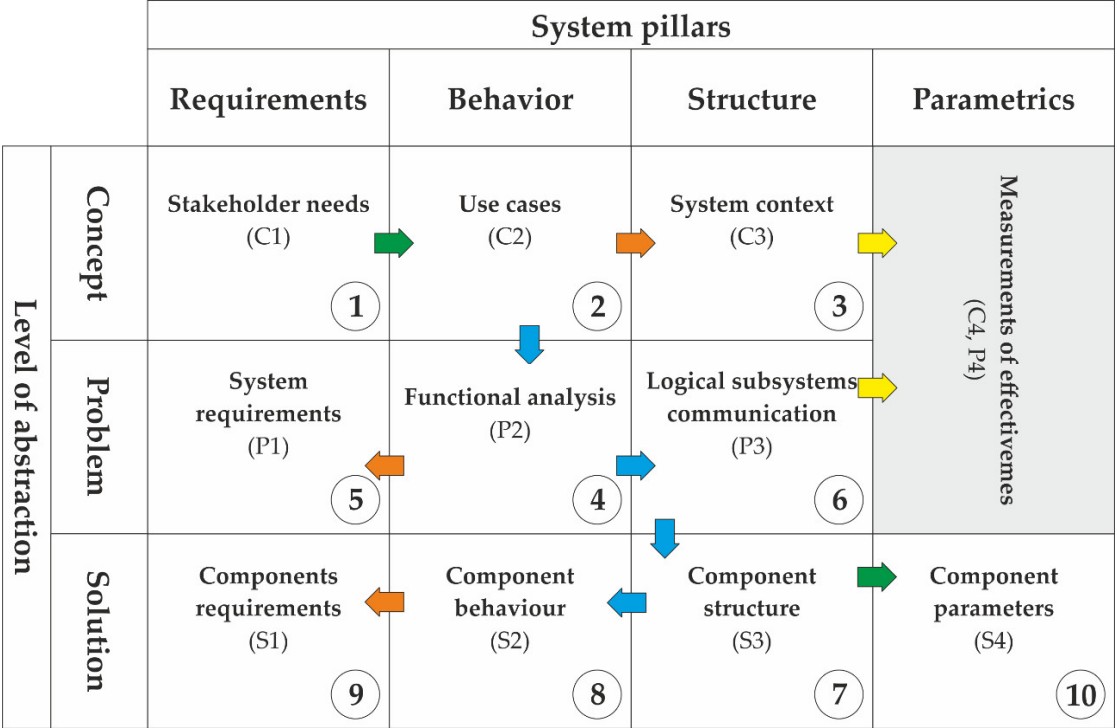

**Figure 4.** *MagicGrid* workflow [33].

Systems modelling with *MagicGrid* is defined with three layers (*Concept, Problem, Solution*) and four system pillars (*Requirements, Behaviour, Structure, Parametrics*). The detail level and completeness of *MagicGrid* allows every team member to precisely share information about their tasks in the process. There is no need to consider the functionality of the entire system as the modularity of the resulting solutions is highly encouraged.

*MagicGrid* workflow found in Figure 4 is further expanded in Figure 5 in the form of activity diagram. It describes the decision flow after adoption of this methodology for complex system development.

At the very beginning, the selected business models (B1) must be described within the set of stakeholder needs (C1). *MBSE* usually results in a modular architecture that supports utilization of different business models for the selected subsystems. Expectations for the product/prototype and the whole project are modelled with the use of *SysML* Requirements Diagram.

The use cases (C2) are defined in an accordance with the client's needs and business goals. It is one of the most important steps in the system modelling process. In this phase, the system engineers define interaction scenarios for users and other systems. This is done regardless of the later implementation. All of the use cases are defined here (C2) no matter what their specificity is, that is, general and detailed ones.

In a typical scenario for most of the mechatronic systems projects, definition of the use case model influences and updates stakeholder needs (C2–C1). When requirements (C1) and use cases (C2) are defined, then the system realization context (C3) can begin.

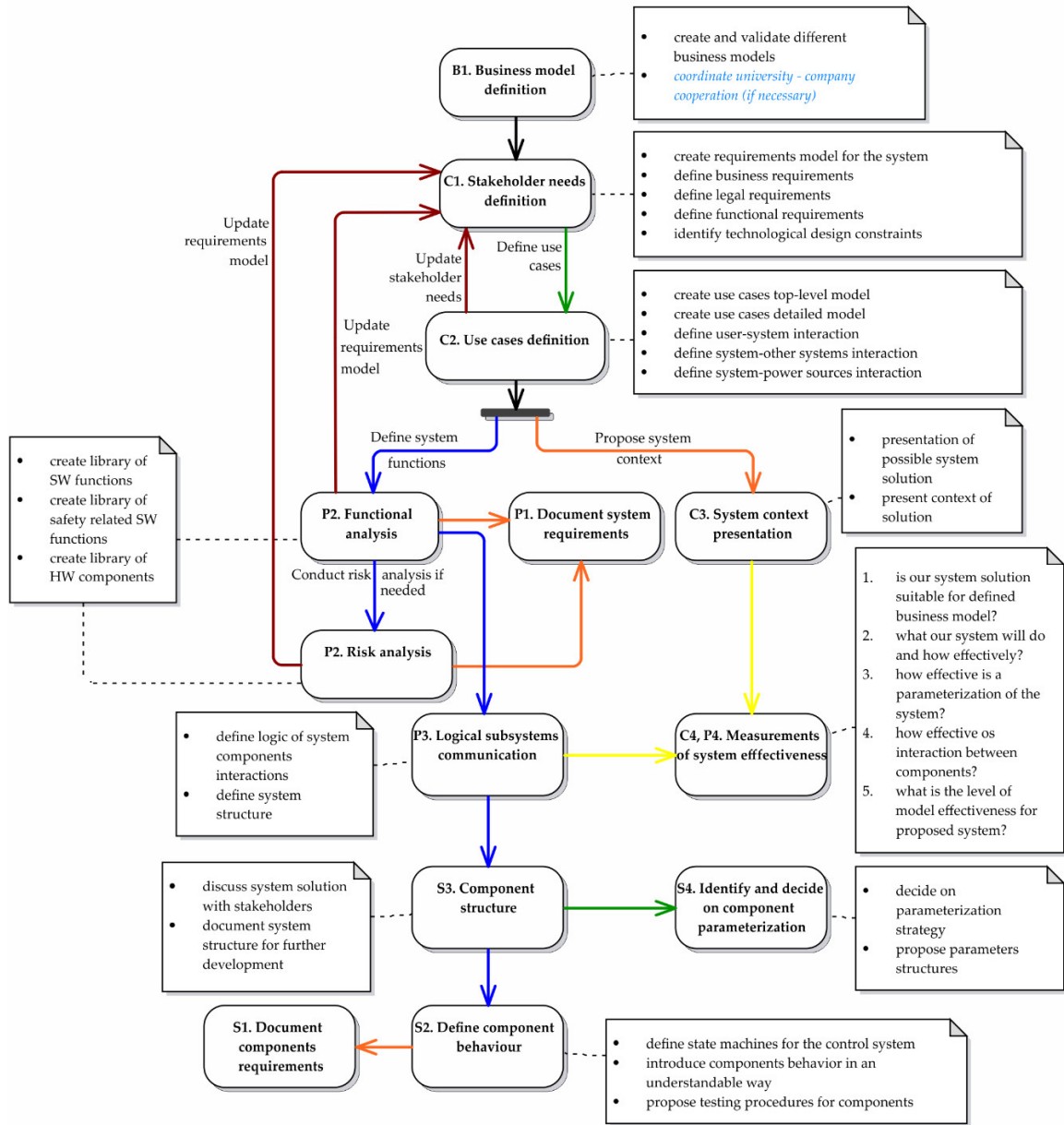

**Figure 5.** *MagicGrid* process in complex system design [33].

Functional analysis (P2) takes place after acceptance of the system context. The use case definition and documentation lets the project team to analyse, identify and summarize the set of functions that needs to be implemented (P2). Risk and functional analysis is crucial when working on the products and must be documented precisely. The procedures and guidelines for these two issues are presented in books [8,9] and standards [34–36]. Information found in these sources is applicable to most of the mechatronic devices available on the market. Risk analysis is heavily supported in software tools for systems modelling.

Functional and risk analysis (P2) will most likely result in an update of the stakeholder needs. After realizing the complexity of the system (stakeholders usually are not familiar with technical aspects of the solution) final clients can drop some of their requirements (C1) or create completely new ones. It can even influence the defined use case model (C2).

Functional analysis (P2), use cases (C2), model of business requirements (C1) all result in formulation of the system requirements (P1). A model for P1 consists of: functional requirements, control hardware constraints, data measurement and its parameters (type, resolution, compliance with

proper standards). After finalizing the P2 phase, the very next step according to Magic Grid (Figure 5) is the definition of the communication logic between subsystems (P3).

Proper documentation of interfaces between components (P3) and optional parametrization (P4) closes the stage of formulating problems and tasks to be solved during the implementation of the final system.

According to *MagicGrid* (Figure 4) workflow, Component Structure (S3) is the first step in the Solution layer. Definition of the components behaviour (S2) and summarizing their requirements (S1) exhausts the set of *MagicGrid* methodology-based activities for system development.

### 2.4. Modelling Tools

Nowadays, there are many software tools for systems modelling and engineering available on the market. The most popular are listed as follows:

- Cameo Systems Modeler (formerly known as: MagicDraw with *SysML* plugin); Vendor: NoMagic; Web: http://www.nomagic.com/products/cameo-systems-modeler.html,
- Innoslate; Vendor: SPEC Innovations; Web: https://www.innoslate.com/,
- Enterprise Architect with SysML plugin;Vendor: SparxSystems; Web: http://www.sparxsystems.com/products/mdg_sysml.html,
- Modelio; Vendor: ModelioSoft; Web: http://www.modeliosoft.com/en/products/solutions/system-architect-solution-overview.html,
- Papyrus 4 SysML, Web: http://www.eclipse.org/modeling/mdt/papyrus/,
- IBM Rhapsody; Vendor: IBM; Web: http://www.ibm.com/software/rational/products/rhapsody/sysarchitect/,
- ARTiSAN Studio; Vendor PTC; Web: http://www.atego.com/products/artisan-studio/.

These tools are the most popular due to their advanced features and support for team-oriented development.

Modelling software architecture and simulation research of new products including control systems is possible due to the features present in the following applications:

- Matlab/Simulink; Vendor: Mathworks,
- MapleSim; Vendor: Maplesoft,
- LMS Imagine.Lab Amesim; Vendor: Siemens,
- Scilab, open source project, initially started by INRIA, France,
- HOPSAN, from Linkoping University of Technology, Sweden,
- OpenModelica, open-source Modelica-based modelling and simulation environment, supported mainly by the Open Source Modelica Consortium (OSMC),
- Dymola.

Most of the integrated development environments (IDE) for industrial control systems [37] compliant with the IEC 61131-3 standard [38] allows validation of the designed algorithms by the means of offline or real-time simulation before it is compiled to the production code.

### 2.5. MBSE in Practice

The bio-inspired robotic hand project [39,40] is a complex mechatronic system. Starting from requirements, through use cases, mechanical and electrical architecture design, up to control algorithm functions, they all can be embedded within the *MagicGrid* process. The system is well-defined with many potential product variants.

The mechatronic design of Reaktor Daya [41] is concerned with mechanical and electrical properties of the system. Selection of materials with sufficient stress resistance is considered and requirements set for the electrical part (power flow, undervoltage) must also be met. Analysis of the

mechanical model (CATIA) for the construction part with ETAP software used for electrical calculations fulfils the *MBSE* paradigm.

Mechatronic design of an extrusion-based additive manufacturing machine is proposed in Reference [42]. In this work, the system design of a complex machine is conducted by integrating several problems (construction design, control system components and control software functions) into a single process. It is a good demonstration of following the *MBSE* workflow: starting from system's requirements, up to their final verification and practical application of the designed device.

## 3. Metamodeling. Domain-specific Modelling Languages

As presented in the previous section, models can be a very powerful tool that describe the structure, behaviour and properties of the devised systems. It is especially useful in the development of control systems for mechatronic devices. Consequently, the increasing presence of modelling paradigms [7] in control system design leads to a situation where defining custom modelling languages is a necessity. The ability to manipulate, analyse or modify the designed models is present in the more sophisticated software tools available on the market.

If the designed models are to be useful as an analytical tool, the underlaying modelling language must support such functionality. This means that the flexibility of design with the use of models heavily depends on the modelling language used (i.e. metamodel). If, for some reason, the currently used modelling tools do not offer such functionality, then one must consider switching to other software solutions that will support the creation of a new modelling language. In this paper, two tools are mentioned that represent current state-of-the-art: Enterprise Architect Ultimate Edition (MDG Technology Builder) and MetaEdit+ 5.5.

### 3.1. Metamodeling Concepts

Metamodels are used to describe the organization and structure of the created models. Meta-metamodels (M3) are a formal description of metamodels (M2); they define the rules applied in the metamodeling process (M2). Metamodel (M2) describes the syntax of models (M1). Additionally, metamodels (M2) define the semantics of models (M1) by the means of extensions and additional rules. This layered approach to metamodeling is shown in Figure 6.

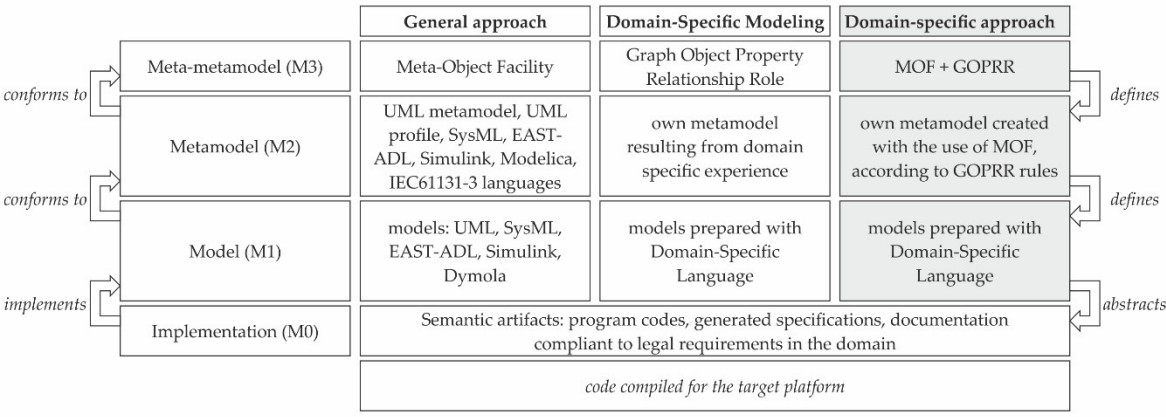

**Figure 6.** Four-layer metamodeling approach. Meta-metamodel (M3) defines abstractions permitted in metamodeling (M2). Metamodel (M2) describes the syntax of models (M1).

In the four-layer approach artefacts in each layer conform to, and/or are abstracted in the higher (i.e. more abstract) layer. For example, semantic artefacts in the M0 layer are abstracted in models from the M1 layer, which in turn conform to the metamodel from the M2 layer:

1.　Meta-metamodel (M3) defines metamodel (M2);
2.　Metamodel (M2) conforms to meta-metamodel (M3);

3. 　　　Metamodel (M2) defines syntax of model (M1);
4. 　　　Model (M1) conforms to metamodel (M2);
5. 　　　Model (M1) abstracts modelling/semantic artefacts (M0);
6. 　　　Modelling artefacts (M0) implement model (M1).

Summarizing, a metamodel (M2) basically can be understood as a model defining the language used to describe models (M1).

### 3.2. Metamodeling Goals

Semantic artefacts (M0) from Figure 6 are the real purpose for utilizing metamodeling in control system design. These are data, programs, files, documents, source codes that have some meaning in another context; they are produced through the design process. For example, when utilizing automatic code generation for a PLC program [27,43], it is generated from high-level software models (M1) to the standardized form of IEC 61131-3 compliant control software codes (M0) [37,38,44].

For the metamodeling approach to have a significant impact on the efficiency of the project design process, some of these M0 artefacts must be a result of the language design process or else the modelling process is nothing more but a sophisticated "living specification and documentation." The MOF approach is described in official specifications [11,45,46]. The approach using meta-metamodel GOPRR was presented by Kelly and Tolvanen [12,47].

The flexibility of metamodeling enables many different approaches to the same problem. The solution should be picked that is best-suited to the team's experience or the problem's context. This is shown in the next section.

### Example

Let us take into consideration the *requirement* stereotype. It will be created among other meta model stereotypes with the use of MDG Technology Builder, as shown in Figure 7.

In the first approach (Figure 7a) the *enumeration* stereotype *req type* with a list of attributes is used for the definition of requirement's type. In this case, the modeler (user of the metamodel) can create unlimited number of *req* model instances. Modelling is constrained by the list of possible requirements' types. On the other hand, an inexperienced engineer can model requirements without having initial knowledge about possible types of requirements—they are built-in into the metamodel.

In the second approach (Figure 7b) there is no explicit definition of *req type*. Instead, a stereotyped extension of metaclass *Class* is created. In this case the modeler can create unlimited number of *req type* instances. One of the *taggedValues* (object properties) references *req* stereotype as a target object for an internal association. Such approach can be used for rapid modelling by the experienced engineers.

The third variant (Figure 7c) is designed for the most experienced modelers. It enables complete freedom by completely separating the requirement from the requirement type. These entities are unrelated, the connection, however, may be made with the use of a new entity (dependency) which can tie them together. Such freedom can prove beneficial in teams with good modelling awareness. On the other hand, such level of flexibility is "specific" for a general use modelling language. In the author's opinion the approach presented in Figure 7c loses its Domain-Specific strengths.

The simple example that was presented here explains which approach to semantics one should select based on the team's experience in *Domain-Specific Languages* (DSL) creation.

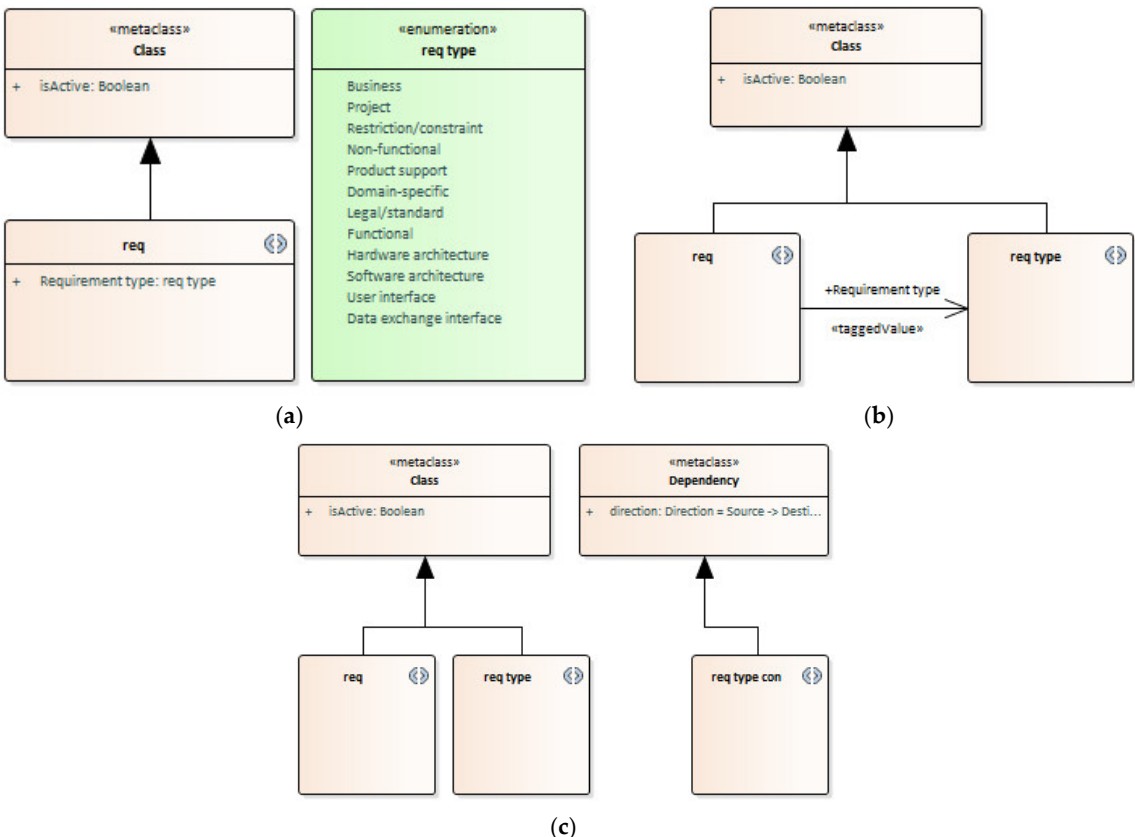

**Figure 7.** Two approaches to the language design. First approach: (**a**) semantic artefacts completely defined at M2 level; Second approach: (**b**) semantic artefacts to be created at level M1 with directed relationship; (**c**) semantic artefacts to be created at level M1 with bi-directional relationship.

### 3.3. Meta-Object Facility. Graph Object Property Relationship Role

The *Meta-Object Facility* Specification (MOF) is the foundation of OMG's industry-standard environment where models can be exported from one application, imported into another, transported across the network, stored in a repository and then retrieved, converted into different formats [45]. Metamodeling layers of MOF are shown in Figure 6.

The *Domain-Specific Language* design proposed by Kelly and Tolvanen [12] utilizes *GOPRR* (*Graph-Object-Property-Relationship-Role*) meta-metamodel. This meta-metamodel allows defining the metamodel in the form of a DSL. Objects as basic DSL elements can be connected with *Relationships* which define a *Role* for the connection to an *Object*. The connection to an Object can be further refined by a *Port* to which the connection is attached. The *Port* is attached to the *Object*, while the *Role* is attached to the *Relationship*. *Objects* and their *Relationships* can be presented in a *Graph*. *Properties* can be added to each of these elements (*Object*, *Relationship*, *Role*, *Port* and *Graph*). This approach is implemented directly in the MetaEdit+ software (www.metacase.com).

Figure 8 presents the workflow for creation of domain modelling languages applied in two software solutions: *Enterprise Architect Ultimate Edition* (MDG Builder) and MetaEdit+ 5.5.

Some analogies can be observed between MOF (MDG Technology) and GOPRR (MetaEdit+):

- MOF: diagram—GOPRR: Graph;
- MOF: stereotype—GOPRR: Object;
- MOF: tagged value—GOPRR: Property;
- MOF: connector—GOPRR: Relationships;
- MOF: role—GOPRR: Role.

In general, these two approaches are identical—the differences stem from the notation used and the market popularity of the modelling software that supports them.

| MOF (Enterprise Architect) | GOPRR (MetaEdit+) |
|---|---|
| Create UML profile, including stereotypes, tagged values, connectors, generators | Create: Objects, Properties, Relationships, Roles, generators |
| Create diagram profile, indicate stereotypes and relationships to be used on a Diagram types | Create: Graph, indicate objects to be used on a Graph type |
| Create diagram toolbox, i.e. icons of the objects to be used on a diagram | Create icons, toolbars for Graphs |
| Compile language with MDG Technology Builder | Export created meta-model for the use in new projects OR create models within the same project (no need to compile of meta-model) |

*Language design stages*

**Figure 8.** Two workflows for metamodeling design process.

### 3.4. Metamodeling as a Stage in Design Process

Overall, the design processes which use the metamodeling approach, for example, *Domain-Specific Languages* for control systems implemented in PLC controllers [48,49], start with determining the domain and its metatypes (stereotypes). They are defined with the use of meta-metamodel (M3) in software solutions dedicated to metamodel (M2) creation.

Figure 9 illustrates the design process of metamodeling. In the case when a final solution (M0) must be modified, it should not be done at the M0 level but rather the M1 models should be modified. After their update, the semantic interpretation should be conducted once more—thus the artefacts (e.g., program code, documentation) are generated again. This process is often called the evolution of the model/application.

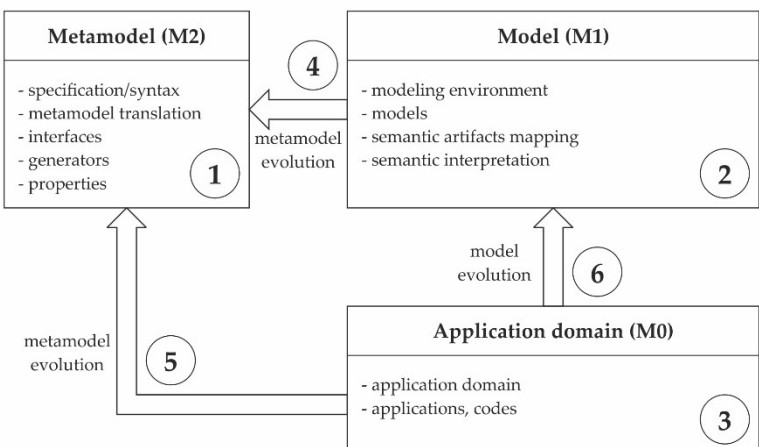

**Figure 9.** Metamodeling design process. (1) metamodeling tools; (2) modelling tools; (3) solution, that is, control system code; (4,5) evolution of metamodel; (6) evolution of model.

If the modelling domain undergoes some changes, for example, caused by the identification of new constraints, new stereotypes or the overall shift in the design philosophy, then the metamodel must be updated (M2). These modifications will be later reflected in the way how the resulting M1 models are designed, thus changing the nature of their implementation.

It can be concluded from Figure 9 that the modification of metamodel (M2) will cause changes in all M1 models; they will be affected by the updated modelling language, both in the newly created elements as well as in the already existing ones.

The metamodel (M2) modification may be caused by analysis of semantic artefacts encountered in the M0 level (number 5 in Figure 9). Their inadequacy may lead to a conclusion that the problem lies in the metamodel itself and that the improvement of these artefacts may be achieved by changes in the M2 layer.

*3.5. Metamodeling Utilization for Control Engineering Projects*

In the case of projects in which new products and processes are designed, metamodeling may be applied in the following areas:

1.  Prototyping of control systems:

    a.  Modelling and rapid prototyping [50,51],
    b.  PLC controller code generation [43,52],
    c.  Dynamic simulation, including generation of simulation models [20,24],
    d.  Target platform programming [23,53,54],
    e.  New algorithm development [55,56];

2.  Systems engineering:

    a.  Systemic approach to design of automation systems (from Business Model Canvas to INCOSE/NoMagic *MagicGrid* systems engineering process) [31],
    b.  Interconnecting business models with research and development [14,57,58],
    c.  Multi-parametric optimization [18,59–62],
    d.  Component-based design [63,64],
    e.  Modularity support for complex systems [65–68];

3.  Standardization:

    a.  Implementation of Machinery Directive/CE, that is, IEC 61508, ISO 13849 [8–10], including risk assessment (ISO 12100) [34],
    b.  Implementation aspects of ISO 26262 [21,69–73],
    c.  Compliance with the IEC 61131-3 standard [37,48,74–79],
    d.  Functional Safety analysis and documentation [9,73,80],
    e.  Implementation of V-model proposed in VDI 2206 [16,81,82],
    f.  Implementation of *EAST-ADL* [59,70,83–87],
    g.  Industry 4.0 architecture metamodel for EU companies cooperation [88–91];

4.  Management:

    a.  Change, project and requirement management [4,15,92,93],
    b.  Product life-cycle management [94,95],
    c.  Research and development team management [96],
    d.  Monitoring of research and development strategy implementation [97],
    e.  Maturity models for Industry 4.0 analysis [98,99].

*3.6. Metamodeling in Engineering Practice*

A very interesting example of project utilizing metamodeling in the *MBSE* workflow is presented in Reference [100–102]. The SGAM (*Smart Grid Architecture Model*) project, realized under the auspices of Smart Grid Coordination Group, resulted in the creation of a domain specific language that is accessible on the webpages of Centre for Secure Energy Informatics (https://www.en-trust.at/downloads/sgam-toolbox/). SGAM Toolbox is an example of European standardization with the use of methods and tools that are presented in this paper.

The authors of SGAM also contributed to the RAMI 4.0 (Reference Architectural Model Industries 4.0) project [88–91], which currently is in an early development stage. RAMI 4.0 metamodel introduces the following modelling layers: *Business*, *Function*, *Informational*, *Communication*, *Integration* and *Asset*. It also proposes its own development process that consists of *Analysis*, *Architecture* and *Design* phases. The modelling philosophy is close to the *Magic Grid* process. RAMI 4.0 is an example of using metamodeling to solve a very complex problem of standardization in modelling intelligent factories for Industry 4.0. Similarly to SGAM, the RAMI 4.0 toolbox was made available and can be downloaded from https://www.en-trust.at/downloads/rami-4-0-toolbox/.

It is also noteworthy that many projects already employing the *MBSE* principles can be further aided in their development by employing the metamodeling approach. One such candidate is the robotic arm discussed at the end of previous section. A dedicated domain-specific modelling language could be used for automatic generation of simulation scenarios for testing different control schemes.

## 4. Mechatronic Systems Under Consideration

In this section a more detailed discussion about the MBSE principles and metamodeling is given that stems from the author's personal experience.

### 4.1. AVIA X-5, 5-axis Machining Centre. New Product Development

Within the X-5 project a new approach for control system design was adopted; it used the concept of mechatronic integration. The machine and its 3D rigid body model developed in SolidWorks is presented in Figure 10.

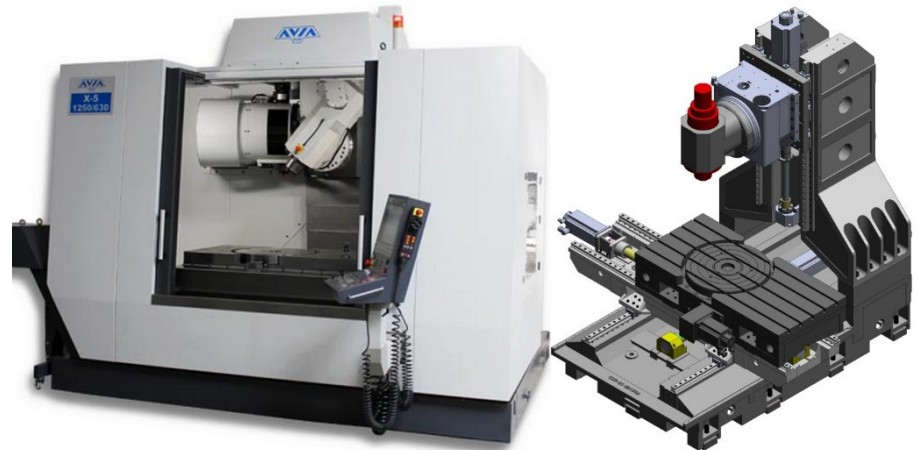

**Figure 10.** AVIA X-5 machining centre and its SolidWorks model.

Prior to modifying the feed-drive module of a 5-axis machining centre (from rotary Permanent Magnet Synchronous Motors, PMSM to Permanent Magnet Linear Motors, PMLM), a full simulation model was automatically generated from the construction assembly model (SolidWorks CAD drawings). Preparation, modification and testing were all conducted in the Simulink environment, thus streamlining the process. The procedure for construction optimization is shown in Figure 11.

It consists of the following steps:

1. A construction assembly of 5-axis milling machine in the form of CAD drawings (M1) is transformed (Simscape Multibody Link, MathWorks, M2 definitions) into a simplified multi-body dynamic model (M1);
2. The simplified multi-body model is extended with models of actuators, friction, dynamics and control. The resulting full Simulink model can be used for virtual verification and validation (M1 level for conducting calculations at M0 level);

3. Optimization follows in which the constraints are considered, that is, dynamic stiffness, geometrical errors, volumetric errors, integral errors; controller's gains are tuned in accordance with the type of motor used (PMSM/PMLM);

4. The construction assembly is modified; different variants of measurements, actuator types are considered. The assembly is updated and the process falls back to Step 2.

The final machine design was achieved after multiple iterations of the optimization process.

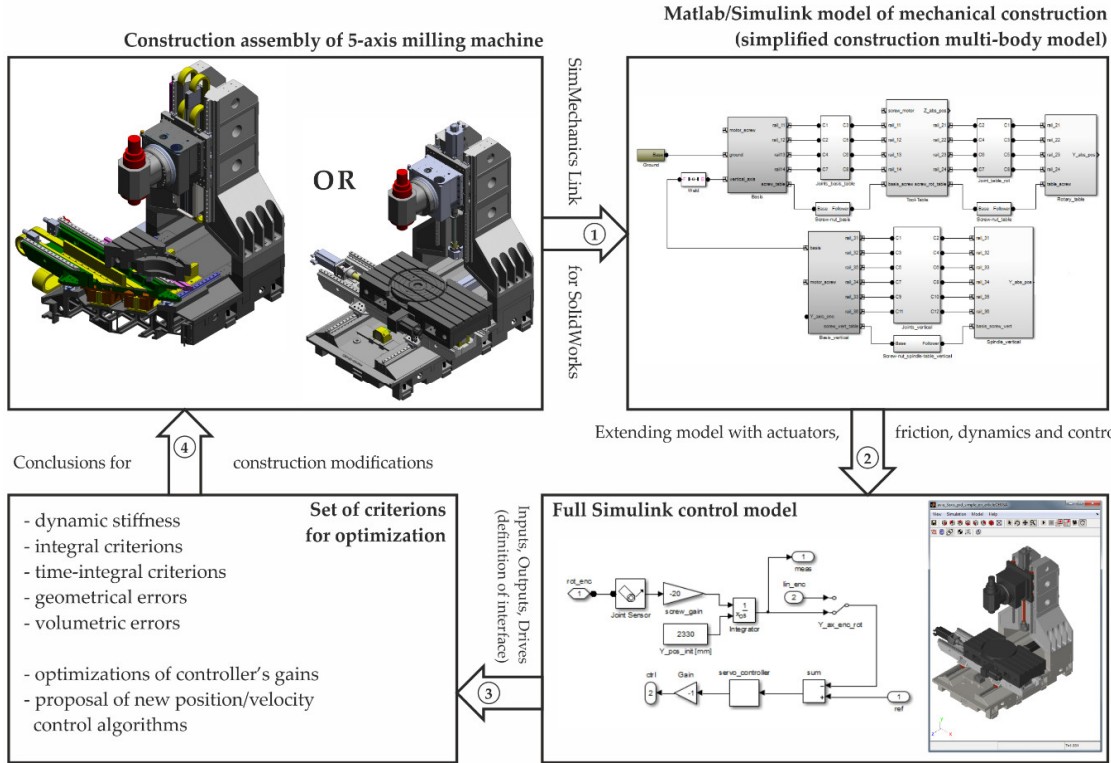

**Figure 11.** AVIA X-5 machining centre virtual optimization workflow.

In the AVIA X-5 project the economic factor was the dominant issue in *Model-Based Design*. The chassis of a milling machine as big as X-5 model takes at least one year to produce (iron casting and later processing, mounting actuators and control system, integration of all of the components). Constructing multiple physical prototypes would be costly and time-consuming. For comparison, three virtual prototypes were created in just three months. This was preceded by one month of team's effort to learn the MBD workflow. The team members' (one automation engineer, two mechanical engineers) prior experience with software tools like SolidWorks (CAD) and Matlab/Simulink (modelling and simulation) certainly shortened this adoption time.

The workflow presented in Figure 11 was proposed by the team leader and was not discussed within the team. At that time (2012), it seemed that it was the best solution to the problem considering the software available to the academic partner. No formal methodology was applied for the project's management.

The MBD approach proved to be very effective for such a small team of engineers—only 3 people were involved at any time. Models of dynamics were created after analysis of data that was acquired by diagnostic features present in iTNC530 control system (Figure 12). This was done for the variant with PMSM motors.

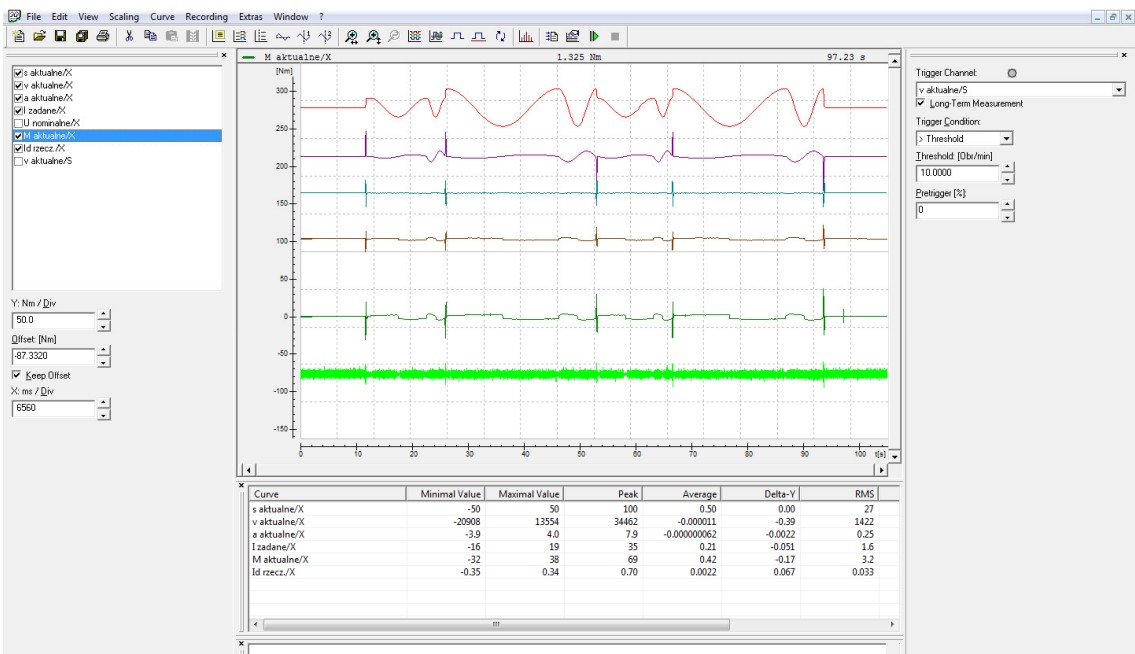

**Figure 12.** TNCScope software screenshot—acquired data signals with the use of diagnostic connection of the iTNC530 control system (Ethernet TCP/IP; 600 μs sampling time).

An optimal solution was found—it was a construction with XY tool table employing linear motors. Four PMLM motors were mounted in parallel in Y axis; another two were put parallelly in X axis. Using the PMLM motors in Z axis was proven to be economically infeasible; instead, PMSM motors were proposed. The final chassis design was too complex and implementing additional safety precautions only increased the already high cost of the machine. However, these measures were necessary as they prevented machine damage in case of power outage.

As already mentioned, due to the approach taken, there was no need to create a physical prototype. Cost to produce one unit was estimated to be 250,000 EUR. Instead, during the project three virtual prototypes were created and tested.

### 4.2. iLOAD Project. Focus on Product and the Process

According to https://cordis.europa.eu/project/rcn/106335/factsheet/en, technological and scientific purpose of the iLOAD project was to meet the needs of civil construction industry and social expectations to decrease the number of occupational hazards caused by cranes. Different industry areas and environmental protection requirements were considered. The main research goal was to develop a new approach to control system in order to improve the operational safety conditions and efficiency of load handling equipment. Another area of interest was the development of new construction approaches to utilize advanced light materials for load handling in order to reduce the weight and fuel consumption of the designed solutions.

These objectives were met thanks to the undertaken multidisciplinary approach to the crane construction. The top-level architecture of control logic was defined and based on the best practices stemming from different industries. Clear and consistent open source architecture enabled outsourcing of various control logic development tasks to specialized companies. Moreover, new structural materials were applied for crane construction based on CFRP (carbon fibre reinforced polymer) and reinforced steel structures. As a result, significant reduction of structure weight was expected.

It is essential to emphasize the role of mutual transfer of knowledge between academic and industrial partners during joint research studies. A synergistic approach was proposed between electronic, mechatronics and materials technologies, based on the knowledge exchanged between the

industrial partners (CARGOTEC NL, PL, SE) strongly involved in the load handling markets and three academic partners (two from Poland, one from Sweden) that had complementary experience.

The project lasted for four years and during this time, an extended training was provided to three novice researchers, five experienced researchers from the partnered institutions and three experienced researchers recruited from outside the consortium. Additionally, a total of 135 months in secondments was planned.

Within "Work Package 1. New control system" the project team was concerned with the development of new way of operating the loader crane, as shown in Figure 13.

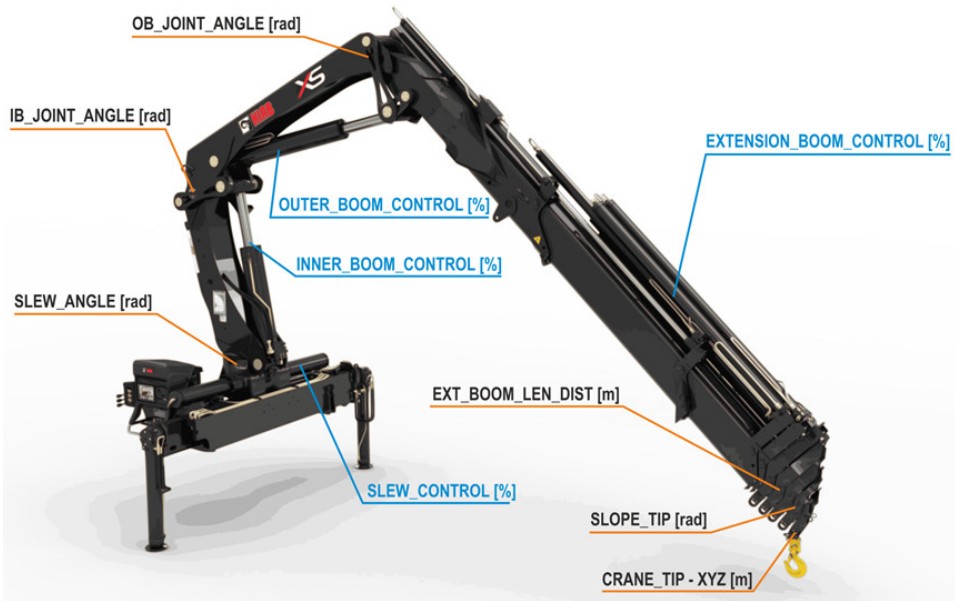

**Figure 13.** HIAB example crane (http://hidrauline-kelimo-technika.enax.lt/hiab-xs-211-pro/).

Normally, such operation is conducted by controlling separate cylinders (SLEW_CONTROL, INNER_BOOM_CONTROL, OUTER_BOOM_CONTROL, EXTENSION_BOOM_CONTROL). Resulting CRANE_TIP – XYZ is normally an output of the system. The proposed approach was to operate in Cartesian space in such a way, that CRANE_TIP – XYZ was a reference value for the control system. This is similar to the Crane Tip Control from HIAB (https://www.youtube.com/watch?v=rpa8VsVuL4Y), extending the approach with automatic SLEW_CONTROL calculation (HIAB solution operates in XZ plane).

Virtual verification and validation test stand prepared during Work Package 1 activities is shown in Figure 14. Due to the project's confidentiality only general view can be published.

Physical user interface (i.e. operator's panel) is used to operate the virtual control system model combined with the simplified crane construction model imported to the Simulink environment from CAD drawings.

Figure 15 depicts one of many tested workflows for control system development. This toolchain combines Control System Design Application (prepared with the use of Qt framework) with custom modelling languages (SpecScop) for system specification. Matlab/Simulink was used for implementation of every system component (hydraulic, mechanical, sensors, digital/logic/safety related control and continuous control). Python scripts were prepared for automatization of CodeSys configuration and PLC programming procedures. For automatic documentation generation regular expressions and Latex were used.

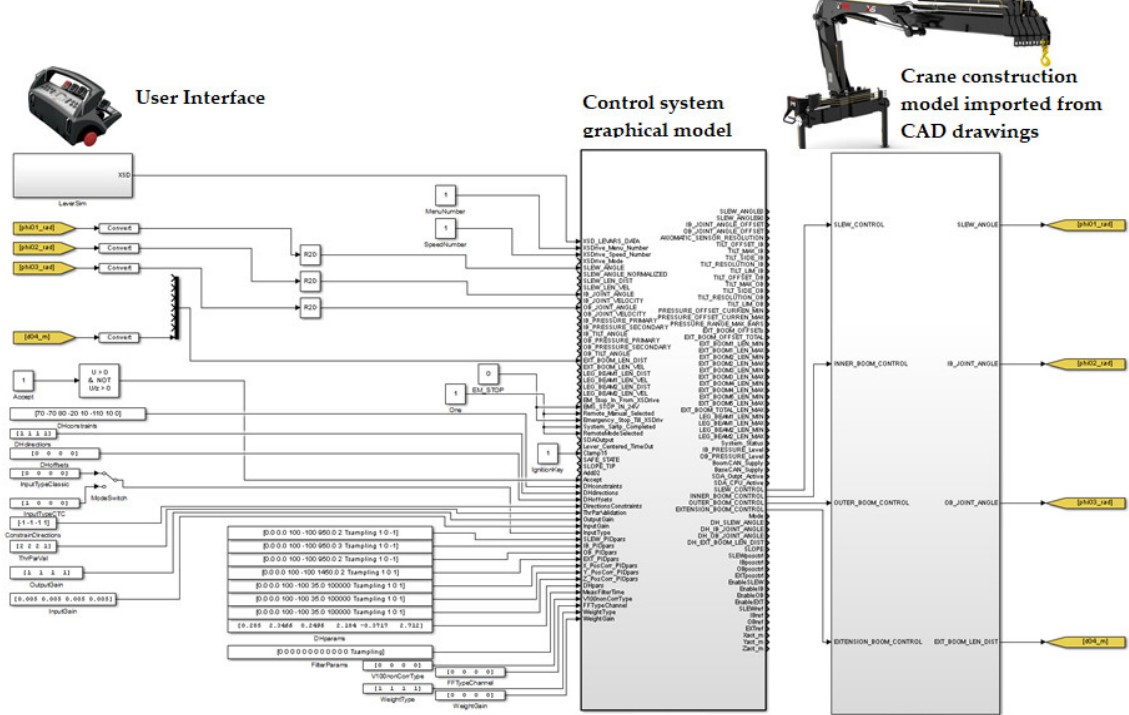

**Figure 14.** Virtual verification and validation of control system.

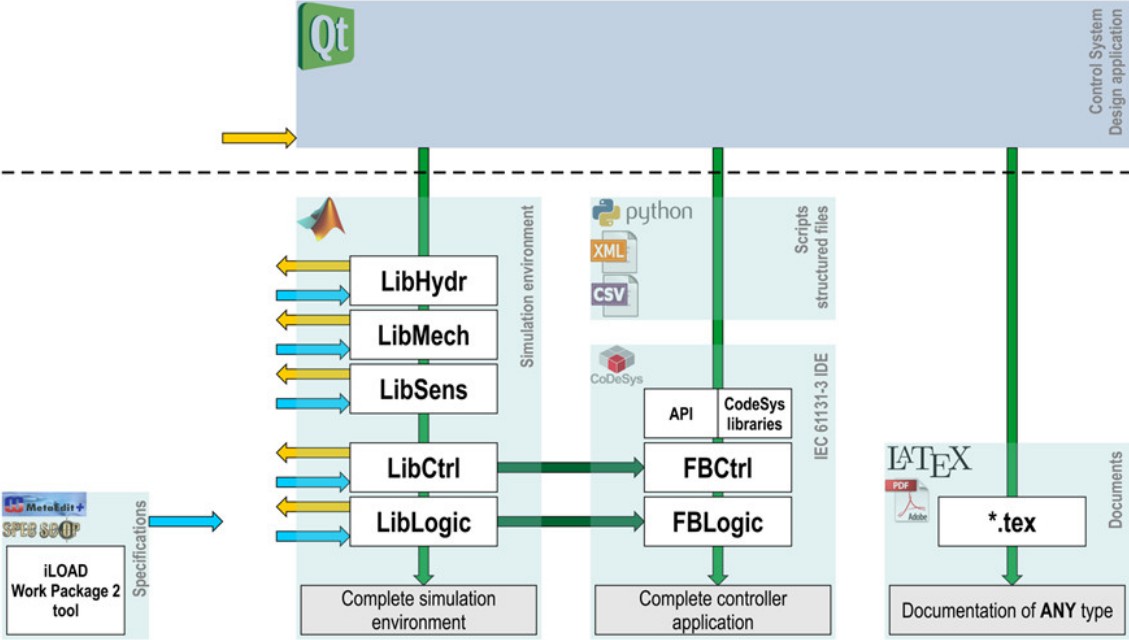

**Figure 15.** One of the toolchains validated at West Pomeranian University of Technology, Szczecin, within the iLOAD project. Multi-vendor software integration: MetaEdit+, Matlab/Simulink, CodeSys v.3.5, Latex/MikTeX, Qt framework, Python.

During Work Package 2 of the iLOAD project, SpecScop metamodel and framework were proposed and prepared with the use of MetaEdit+ software. The main purpose of this simple *Domain-Specific Language* was to hasten the development of Matlab/Simulink libraries by team member's not experienced with Mathworks software tools.

In the iLOAD project, the most important MBD features present during project development, process development and their management, were:

1.  Product aspect—prototyping of new control system functions for loading task without the need to build costly physical prototypes,
2.  Process aspect—possibility of quick reconfiguration of a chosen control system element. This demand was caused by the ever-changing or yet unpublished, operational safety regulations,
3.  Management aspect—improvements in monitoring and management of EU projects. A necessity, considering the rotation of team members.

The MBD approach in iLOAD project was used by a team with 16 members. At the beginning, only three of the participants had any prior experience or were trained in the areas of MBD, DSL and MBSE. However, for the rest of the team, experience from previous projects [22,85–87,103,104] was proven to be invaluable. After 12 months, all of the participants (i.e. institutional and individual) were familiar with intricacies of the *Model-Based Design* approach in their area of expertise:

1.  Product Aspect—3 team members
2.  Process Aspect—8 team members
3.  Management Aspect—4 team members
4.  Full *MBD* knowledge—3 team members

Applying the *Model-Based Design* to new areas was preceded by mini-projects; they served as an introduction to rules, methodology and tools of MBD. Domain specific languages were also designed as a consequence of these projects.

During product development the workflow presented in Figure 15 was picked. It was chosen from many other variants as the most attractive from an academic standpoint and also offered the most robust framework. It would require (if it was implemented, that is) many years of prior work before its full implementation in the industry context. With the addition of new features in PLC/PAC IDEs, commercial involvement in developing custom tools for automatic generation of artefacts (e.g., hardware configuration, software configuration) is not that attractive anymore. Producers of PLC/PAC systems offer fully-featured solutions for their platforms [23].

As a result of applying the MBD approach at MBD-9 level (Figure 1), target goals for the project were reached:

1.  New control features lead to 30% faster adaptation of the designed product by inexperienced operators—the learning curve was successfully reduced. A test scenario was specially designed to verify this target goal;
2.  Movement precision was greatly improved in comparison to the previously applied control system;
3.  Modular architecture of the control system lead to parallel development of system's algo rithms. This resulted in a faster implementation of new control functions.

Applying the *Model-Based Design* approach paved a new way of cooperation between the industrial and academia partners. It was later continued in the ARControl initiative "Application of expanded reality, interactive systems and voice operator interface in control of lifting devices" and was financed under grant agreement no. 245598, Innotech, National Centre of Research and Development, Poland, 2015–2017. Additionally, a scope of minimal knowledge, competence and skills was defined in order to realize projects in the *Model-Based Design* paradigm. It was concluded that the universities' curricula were inadequate and this sentiment was expressed by all the participating team members of different nationalities and backgrounds. It was a very important experience for the author and made him refocus his didactic approach to teaching automatics and robotics.

### 4.3. Modelling and Metamodeling Aspects in the Presented Projects

The projects that were presented in this section were realized with best practices described in the VDI 2206 document [81]; they were also developed in accordance to the *MagicGrid* process (Figures 4 and 5).

In the case of AVIA X-5 project, much of the focus was devoted to analysis and modelling of the machine's control system for movement control (Functional analysis P2, Component behaviour S2, Component structure S3).

In the iLOAD project the shift was made towards integration of the entire design process, from the requirements to the final solution: Stakeholder needs (C1), Use cases (C2), System context (C3), Functional analysis (P2), Component behaviour (S2), Component structure (S3), Component parameters (S4).

In Tables 1 and 2, the selected aspects of modelling, metamodeling and chosen MBD features are presented for both projects.

The main conclusion of the data presented in Table 1 is that requirements engineering and monitoring should be considered regardless of the problem that is solved with the use of *Model-Based Systems Engineering*. *Domain-Specific Languages* must include this important part of system design.

In Table 2, the most important aspects of metamodeling used in AVIA X-5 and iLOAD projects are presented. For each project a different toolchain was prepared:

1. AVIA X-5 project (Figure 11),

    a. SolidWorks CAD drawings (M1) converted with Simscape Multibody Link (M2) into Simulink models (M1);
    b. Simulink models (M1) executed during dynamic simulations (M0);
    c. Documents/reports (M0) generated out of the Simulink models (M1);
    d. Results from calculations (M0) used for manual update of the construction model (M1);

2. iLOAD project (Figure 15),

    a. Modelling language SpecScop (M2) used for specification of system components (M1);
    b. Automatic Matlab/Simulink generation (M2) of simulation models (M1) from specifications (M1); functionality implemented directly in Matlab/Simulink;
    c. Libraries (M1) of components created in Matlab/Simulink are automatically updated (M2) with new specifications (M1);
    d. Control system code (M0) generated from models (M1);
    e. Dynamic simulation (M0) conducted with predefined, parametrized models (M1);
    f. General/detailed documentation (M0) automatically generated from models (M1).

**Table 1.** Modelling aspects.

| Modelling Aspects | AVIA X-5 Project | *iLOAD* Project |
|---|---|---|
| Control system prototyping and development | -dynamic simulation of motion control algorithms, -parametric identification, -modal analysis of machine construction, -dynamic simulation of proposed measurement architectures, -dynamic simulation of PMSM/PMLM motor actuators | -dynamic simulation: hydraulic components, mechanical design, control platform, control algorithm, new sensor technologies, new measurement schemes, new materials, -dynamic simulation of loader cane features, including safety related algorithms |
| Systems and requirements engineering | -hardware/software architecture optimization, -virtual verification of construction prototypes, -closed-loop simulation, -simplified textual requirements (Excel sheets) | -hardware architecture validation, -virtual verification and validation, -graphical specification, -hardware-in-the-loop simulation, -closed-loop simulation, -systematic requirements engineering |

**Table 1.** *Cont.*

| Modelling Aspects | AVIA X-5 Project | iLOAD Project |
|---|---|---|
| Standardization and standard compliance | -IEC 61508. Functional safety of electrical/electronic/programmable electronic safety-related systems, Parts 1-7, <br> -IEC 61131-3:2013. Programmable controllers. Part 3: Programming languages, <br> -ISO/IEC/ IEEE 29148:2011 – textual requirements (template) | -IEC 61508. Functional safety of electrical/electronic/programmable electronic safety-related systems, Parts 1–7, <br> -IEC 61131-3:2013. Programmable controllers. Part 3: Programming languages, <br> -EN 12999:2011+A1. Cranes—Loader cranes, <br> -ISO 13849-2:2012. Safety of machinery—Safety-related parts of control systems—Part 2: Validation, <br> -ISO 15442:2005. Cranes—Safety requirements for loader cranes, <br> -IEC 62061:2005. Safety of machinery—Functional safety of safety-related electrical, electronic and programmable electronic control systems, <br> -VDI—Association of German Engineers VDI 2206—Design methodology for mechatronic systems. Design 2004, 118, <br> -own domain-specific modelling language (EAST-ADL inspired) |
| Project management support | -industry—university cooperation, <br> -factory testing planning (new for university members) | -industry—university cooperation, <br> -project resources planning, <br> -EU Participant Portal reporting, <br> -SCRUM project management, <br> -certified laboratory testing, <br> -project budget monitoring, <br> -team members training planning, <br> -time-sheets monitoring |

In the AVIA X-5 project the toolchain was quite expensive and offered predefined metamodels and ready-to-use generators. Flexibility of development was constrained by the project's budget. Within the iLOAD project a custom toolchain was designed, developed and later used for achieving project goals at each stage. It was much more time-consuming to prepare the toolchain's modules but the resulting flexibility was greater.

**Table 2.** Metamodeling aspects.

| Metamodeling Aspects | AVIA X-5 Project | iLOAD Project |
|---|---|---|
| Control system prototyping and development | -Matlab/Simulink, <br> -Siemens/Haidenhain software tools, <br> -CAD drawings (SolidWorks) transformation into dynamic models, <br> -reports generation, <br> -existing metamodels | -Matlab/Simulink, <br> -MetaEdit+, <br> -automatic Simulink models generation, <br> -PLC code generation, <br> -creation of new metamodels |
| Systems and requirements engineering | -Matlab/Simulink tools, <br> -existing metamodels | -*EAST-ADL, SysML*, <br> -creation of new metamodels, <br> -MetaEdit+ requirements import/export, <br> -Matlab/Simulink, <br> -Latex, Microsoft Office integration |
| Standardization and standard compliance | -Matlab/Simulink, <br> -textual management, <br> -existing metamodels | -creation of new metamodels, *SysML*, <br> -Microsoft Office integration, <br> -import/export of model objects |
| Project management support | -Microsoft Office, <br> -no methodology used, <br> -existing metamodels | -creation of new metamodels, <br> -Microsoft Office integration, <br> -SCRUM tools and process, <br> -Enterprise Architect |

Table 3 presents the most important factors related to teams that participated in each of the projects.

**Table 3.** Team-related aspects of the realized projects

| Team-Related Aspects | AVIA X-5 Project | *iLOAD* Project |
| --- | --- | --- |
| Experience | -Matlab/Simulink, <br> -SolidWorks (CAD), <br> -project participation/management, <br> -control system design, <br> -mechanical system design | -Matlab/Simulink, HOPSAN, <br> -CAD software, <br> -Domain-Specific Modelling, <br> -modelling of production processes, <br> -project participation/management |
| Size | 3 engineers | 16 engineers, incl. 3 managers |
| Expertise | -control engineering, <br> -mechanical engineering | -control engineering, robotics, <br> -mechanical engineering, <br> -hydraulic engineering, <br> -IT, ICT, <br> -project management, <br> -standardization, functional safety |
| Motivation/rationale | -business needs, <br> -industry – university cooperation, <br> -new machine/product development | -business needs, <br> -industry—university cooperation, <br> -new machine/product development, <br> -new process, <br> -standards compliance, <br> -new people, <br> -new materials |
| Method adoption time | 1 month | 12 months |

## 5. Discussion

From the point of view of MBD and MBSE approaches, the most valuable part of the presented projects was the integration of engineers with different backgrounds and transfer of knowledge that followed.

The projects presented in the third section opened the discussion on generic problems of metamodeling utilization for development of mechatronic and cyber-physical systems. The methods proposed in this section can improve the quality and efficiency of these processes.

Table 1 which summarizes the modelling effort in AVIA X-5 and iLOAD projects leads to the following conclusions:

1. Control systems prototyping is a very important aspect in machine design. The same can be said about simulations conducted in a closed loop. Essential information is gathered that can help to evaluate the feasibility of the proposed design.
2. Requirements engineering is essential for testing the product both in simulations and in a real environment.
3. The rationale for employing the *Model-Based Design* increases with the number of requirements and standards that the product must adhere to.
4. *Model-Based Design* is especially useful in projects which involve different participating parties. In the case of the projects presented in the previous section, the cooperation involved representatives from industry and academia. It benefitted everyone and resulted in deeper understanding of the encountered problems.

Analysis of Table 2 yields the following observations:

1. Using metamodels that are already built-in the utilized tools (AVIA X-5 Project) allows to solve complex problems without the need to modify the modelling process.
2. Possibility of metamodel modification and creation of custom generators (iLOAD Project) opens limitless possibilities in the area of project process creation. However, this is only possible with personnel that is experienced in the field of model design. These engineers are called metamodellers.

3.  Utilization of commercially available tools for modelling and requirement management is quickly becoming burdensome for the users. The offered functionality soon becomes inadequate to fully describe the modelled process.
4.  Parametrization and multi-criterion optimization of systems with multi-source requirements is possible only when custom-made modelling languages are created; these languages must consider the requirements' parameters and relations between them and systems' components or documentation.
5.  Metamodeling allows to apply a hybrid approach to project management. This means interspersing heuristic and standard methods within one management process.

Table 3 provides information for teams that are interested in implementing the *Model-Based Design* in their projects:

1.  Several years of experience in a given problem domain is required in order to properly apply *Model-Based Design*. The benefits of the MBD approach cannot be fully appreciated without complete knowledge of the field in which it is applied.
2.  Full knowledge of all MBD aspects is not required when it is adopted for the first time—simultaneous and effective implementation of its every feature is impossible.
3.  Some aspects of *Model-Based Design* can be used even in small groups; with larger teams the payoff is even bigger as more MBD elements are used.
4.  Motivation for applying *Model-Based Design* in mechatronic and cyber-physical systems is often independent from the product itself. It can be a result of research and development projects in which industry and academia partners cooperate to create new products or processes. This is an unusual order-of-business for both parties, additionally limited by the specifics of research program and its funding.
5.  The issues caused by changes in team's personnel also prove the *Model-Based Design* to be beneficial. It enables seamless introduction of new employees into the project and enables safe continuation whenever a more experienced team member leaves.
6.  *Model-Based Design* improves communication between the stakeholders and the project team that designs the system in question.

*Generic Workflow for Model-Based Design*

The methods proposed in this subsection can improve the quality and efficiency of mechatronic and cyber-physical systems design processes. Regardless of the problem's complexity (new dexterous robotic hand [39,40], experimental reactor [41], extrusion-based additive manufacturing machine [42], satellite [20,24,25], loader crane [105], 5-axis milling machine [61]), the following workflow can be applied:

1.  Introduce *MagicGrid* process (including VDI 2206 [81] V-model approach at each stage, no matter if modelling requirements, use cases, system architecture or functions),
2.  Define *Domain-Specific Language* use cases for the project,
3.  Propose initial version of the language; model part of the system as a pilot sub-project,
4.  Update modelling language, define abstraction levels/layers for modelling,
5.  Define scope for code/documentation generation,
6.  Define scope for data import/export and select candidate objects,
7.  Model most of the system and close code/documentation generation loops,
8.  Integrate the proposed modelling language with project management methodology, for example, *SCRUM*, *Project Cycle Management* or propose custom modelling process.

This workflow provides a generic approach for complex (mechatronic and cyber-physical [82,106–108]) system design. One can combine General Purpose Modelling Languages (*UML*, *SysML* [5]) with

*Domain-Specific Languages* within a single ecosystem. Using only general-purpose languages is far less effective even when a precise methodology (INCOSE or *MagicGrid*) is applied.

Metamodeling of *Domain-Specific Languages* strongly supports:

1. control system development, including development of new control functions, components selection, dynamic simulation and code generation,
2. systems engineering, including virtual verification and validation, hardware in the loop simulation, requirements engineering at each stage of the project; integration of multicriteria indices (similar to the proposed in Reference [62] mechatronic design quotient MDQ) for ensuring high quality of the resulting system,
3. project management: higher quality of decision-making process achieved by rapid analysis of the project's state.

During work on modelling languages in the mechatronic projects presented in this paper, the following problems were identified:

1. Utilization of General Purpose Languages (*UML*, *SysML*) takes a lot of time compared with *Domain-Specific Languages*. This is true even for languages that take basic concepts from *SysML* (like Requirement or Use Case). This conclusion is similar to [109],
2. Issues were observed with engineers that observe their increasing efficiency—they are exhibiting problematic social behaviour that stems from the increased quality and efficiency. This is a common problem in digitalization projects, when engineers are forced to use new toolchains which are usually far more efficient than the currently used. This problem, however, is outside the scope of this paper and needs further research.

Experience collected from over 20 research projects (including AVIA X-5 and iLOAD) in which selected elements of *Model-Based Design* were applied, culminated in a decision in 2014 to develop a modelling language that would support effective project management. The main goal was to intertwine a heuristic methodology with a well-rounded, popular counterpart that is often used in software development projects, including embedded systems [110].

The result of this effort is the *researchML* language. The language extends processes like *Magic Grid* or INCOSE by expanding *SysML* and attempts to improve management of companies and research projects. Its structure offers a manageable way to track the team's qualities and current workload. Moreover, it can be used to quickly build project portfolios for potential investors. It is the main scientific contribution of this paper.

## 6. ResearchML

*ResearchML* was developed in 2016 and was a response to the increasing need for organizing the procedures for management of purely scientific and R&D projects developed by the team working in Mechatronics Centre at West Pomeranian University of Technology, Szczecin. The type of activities pursued by project members varied greatly, partly due to differences in research interests, experience and broad scope of the realized projects. These factors posed an increasing difficulty in managing such a varied team and hindered the realization of projects.

Additionally, the legislation changes in Poland and demographic decline heavily influenced the team's efficiency and quality of work. The team conducted multiple projects in 2006–2016 aimed at improving the control systems of CNC machines. The increase in the number of projects and, consequently, the number of tasks each team member was responsible for, had led to a conclusion that a modelling language was needed in order to improve the team's management.

### 6.1. ResearchML Goals

At first, during the *researchML* development a set of goals was defined for the construction and application of the modelling language. These were described below.

### 6.1.1. Activity Management and Effective Reporting

The work time of research teams that are involved in a broad selection of problems across multiple projects has to be monitored. This is achieved by implementing time-sheet systems which allow to monitor the time spent on each project and activities within. The reports are generated on a monthly or quarterly basis, the latter type is tied to the financing of project phases. The complexity of this issue is increased by the fact that the team members' contributions vary across the project's realization.

### 6.1.2. Information Flow Optimization

Whenever the involvement in a design process includes representatives from industry and academia or whenever the partners are spread around the world, the use of graphical modelling languages shared as a network resource (i.e. repositories) enables an efficient communication.

### 6.1.3. Fast Adaptation

Often, during the project's realization new members join the team; these engineers must work in areas in which the groundwork was not laid by them. Static documents cause long adaptation times and endanger the deadlines. The use of *Domain-Specific Languages* causes the minimalization of that transition period.

### 6.1.4. Compensation for Lack of Managerial Experience

In control system design the technological knowledge is of utmost importance. This often leads to team leaders that do not possess enough managerial experience. Conversely, the opposite also holds—the technological competence of a person well predisposed to lead a group of people may be inadequate. The answer to the first concern could be the development of a *Domain-Specific Language* that should reinforce the engineers with high technical prowess by supplying them with tools aiding the project management. An example of such language is *researchML*.

The requirements set for the modelling language caused that the devised metamodel found the following applications:

- reporting,
- preparing grant applications,
- storage and organization of project's information,
- project management,
- management and organization of research infrastructure,
- monitoring workload of each project member,
- career development plan for employees, training plans for acquiring new skills.

### *6.2. ResearchML Design*

Abstraction layers for the actual version (3.09) of *researchML* language are shown in Figure 16 (with stereotypes):

1. Project Analysis Level (*address*, *project*, *workplace*),
2. Project Design Level, Functional Project Design (*goal*, *goal type*, *project phase*, *project implementation rate*, *project rate type*, *project rate value*, *project role*, *project task*, *researcher*, *risk*, *risk type*, *workload*, *workload set*),
3. Project Design Level, Resources Project Design (*author*, *cost*, *cost type*, *experience*, *experience type*, *resource*, *resource type*, *service*, *service group*, *technology*, *technology category*),
4. Project Implementation Level (*requirement*, *relationship*),
5. Project Implementation Level, Relationships (*contain*, *copy*, *derive*, *satisfy*, *trace*, *verify*),
6. Project Operational Level (*project activity*, *project activity type*, *task activity*, *task activity type*).

List of objects, relationships and target object roles is presented in Figure 17. It is preferable for relationships within abstraction levels to be contained in metamodel and in objects (stereotypes). In this case the creation of generators is straightforward. Relationships going beyond the abstraction levels should be modelled at the M1 level—the resulting language and generated models are then easier to implement.

### 6.2.1. Project Analysis Level

This level contains general information about the realized project (current stereotypes: *project*, *address*, *workplace*), its name and description (*description*). An institutional project leader is specified (*Project leader* in *workspace* stereotype, additionally the GPS localization is supplied). As the project members locations are geographically distinct, the metamodel contains the information about their whereabouts. This is realized by the *address* stereotype which contains the following: country, city, postal address. One workspace is assigned one address. Similarly, a single project can have only one institutional project leader.

| Source \ Target | address | project | workplace | goal | goal type | project phase | project impl. rate | project rate type | project rate value | project role | project task | researcher | risk | risk type | workload | workload set | author | cost | cost type | experience | experience type | resource | resource type | service | service group | technology | technology cat. | requirement | contain | copy | derive | satisfy | trace | verify | relationship | project activity | project act. type | task activity | task act. type |
|---|---|---|---|---|---|---|---|---|---|---|---|---|---|---|---|---|---|---|---|---|---|---|---|---|---|---|---|---|---|---|---|---|---|---|---|---|---|---|---|
| **Project Analysis Level** | | | | | | | | | | | | | | | | | | | | | | | | | | | | | | | | | | | | | | | |
| address | | | | | | | | | | | | | | | | | | | | | | | | | | | | | | | | | | | | | | | |
| project | | | 1 | | | | | | | | | 2 | | | | | | | | | | | | | | | | | | | | | | | | | | | |
| workplace | 3 | | | | | | | | | | | | | | | | | | | | | | | | | | | | | | | | | | | | | | |
| **Project Design Level, Functional Project Design** | | | | | | | | | | | | | | | | | | | | | | | | | | | | | | | | | | | | | | | |
| goal | | | | | 4 | | | | | | | | | | | | | | | | | | | | | | | | | | | | | | | | | | |
| goal type | | | | | | | | | | | | | | | | | | | | | | | | | | | | | | | | | | | | | | | |
| project phase | | 5 | | | 6 | | | | | | | 7 | 8 | | | | | | | | | | 9 | | | | | | | | | | | | | | | | |
| project impl. rate | | 10 | | | | | 11 | 12 | | | | | | | | | | | | | | | | | | | | | | | | | | | | | | | |
| project rate type | | | | | | | | | | | | | | | | | | | | | | | | | | | | | | | | | | | | | | | |
| project rate value | | | | | | | | | | | | | | | | | | | | | | | | | | | | | | | | | | | | | | | |
| project role | | | | | | | | | | | | | | | | | | | | | | | | | | | | | | | | | | | | | | | |
| project task | | 13 | | | | 14 | | | | | | 15 | | | | | | | | | | | 16 | | | | | | | | | | | | | | | | |
| researcher | | 17 | 18 | | | | | | | 19 | | | | | 20 | 21 | | | | | | | | | | | | | | | | | | | | | | | |
| risk | | | | | | | | | | | | | 22 | | | | 23 | | | | | | | | | | | | | | | | | | | | | | |
| risk type | | | | | | | | | | | | | | | | | | | | | | | | | | | | | | | | | | | | | | | |
| workload | | 24 | | | | 25 | | | | | | 26 | | | | | | | | | | | | | | | | | | | | | | | | | | | |
| workload set | | | | | | | | | | | | | | | | 27 | | | | | | | | | | | | | | | | | | | | | | | |
| **Project Design Level, Resources Project Design** | | | | | | | | | | | | | | | | | | | | | | | | | | | | | | | | | | | | | | | |
| author | 28 | | | | | | | | | | | 29 | | | | | | | | | | | | | | | | | | | | | | | | | | | |
| cost | | | | | | | | | | | | | | | | | | | 30 | | | | | | | | | | | | | | | | | | | | |
| cost type | | | | | | | | | | | | | | | | | | | | | | | | | | | | | | | | | | | | | | | |
| experience | | | | | | | | | | | | | | | | | 31 | | | | 32 | | | | | | | | | | | | | | | | | | |
| experience type | | | | | | | | | | | | | | | | | | | | | | | | | | | | | | | | | | | | | | | |
| resource | | | | | | | | | | | | | | | | | 33 | | | | | | 34 | | | | | | | | | | | | | | | | |
| resource type | | | | | | | | | | | | | | | | | | | | | | | | | | | | | | | | | | | | | | | |
| service | | | | | | | | | | | | | | | | | 35 | | | | | | | | | 36 | | | | | | | | | | | | | |
| service group | | | | | | | | | | | | | | | | | | | | | | | | | | | | | | | | | | | | | | | |
| technology | | | 37 | | | | | | | | | | | | | | 38 | | | | 39 | 40 | 41 | | | | 42 | 43 | | | | | | | | | | | |
| technology cat. | | | | | | | | | | | | | | | | | | | | | | | | | | | | | | | | | | | | | | | |
| **Project Implementation Level, Relationships** | | | | | | | | | | | | | | | | | | | | | | | | | | | | | | | | | | | | | | | |
| requirement | | | | | | | | | | | | 44 | | | | | 45 | | | | | | | | | | | | | | | | | | | | | | |
| contain | | | | | | | | | | | | | | | | | | | | | | | | | | | | | | | | | | | 46 | | | | |
| copy | | | | | | | | | | | | | | | | | | | | | | | | | | | | | | | | | | | 47 | | | | |
| derive | | | | | | | | | | | | | | | | | | | | | | | | | | | | | | | | | | | 48 | | | | |
| satisfy | | | | | | | | | | | | | | | | | | | | | | | | | | | | | | | | | | | 49 | | | | |
| trace | | | | | | | | | | | | | | | | | | | | | | | | | | | | | | | | | | | 50 | | | | |
| verify | | | | | | | | | | | | | | | | | | | | | | | | | | | | | | | | | | | 51 | | | | |
| relationship | | | | | | | | | | | | | | | | | | | | | | | | | | | | | | | | | | | | | | | |
| **Project Operational Level** | | | | | | | | | | | | | | | | | | | | | | | | | | | | | | | | | | | | | | | |
| project activity | | | | | | | | | | | | 52 | 53 | | | | | | | | | | | | | | | | | | | | | | | 54 | | | |
| project act. type | | | | | | | | | | | | | | | | | | | | | | | | | | | | | | | | | | | | | | | |
| task activity | | | | | | | | | | | | | | | | | | | | | | | | | | | | | | | | | | | | 55 | | 56 | |
| task act. type | | | | | | | | | | | | | | | | | | | | | | | | | | | | | | | | | | | | | | | |

**Figure 16.** *ResearchML* 3.09 abstraction layers and objects.

| No | Abstraction Level | Source | Relationship | Target role | Target |
|---|---|---|---|---|---|
| 1 | Project Analysis Level | project | ... | Project leader | workplace |
| 2 | | project | ... | Project manager | researcher |
| 2 | | project | ... | Project R&D manager | researcher |
| 3 | | workplace | ... | Address | address |
| 4 | Project Design Level, Functional Project Design | goal | ... | Goal type | goal type |
| 5 | | project phase | ... | Project | project |
| 6 | | project phase | ... | Phase effects | goal |
| 7 | | project phase | ... | Researchers | researcher |
| 8 | | project phase | ... | Phase risk | risk |
| 9 | | project phase | ... | Phase resources | resource |
| 10 | | project implementation rate | ... | Project | project |
| 11 | | project implementation rate | ... | Rate type | project rate type |
| 12 | | project implementation rate | ... | Base value | project rate value |
| 12 | | project implementation rate | ... | Target value | project rate value |
| 13 | | project task | ... | Project | project |
| 14 | | project task | ... | Project phase | project phase |
| 15 | | project task | ... | Project team | researcher |
| 16 | | project task | ... | Task resources | resource |
| 17 | | researcher | ... | Project | project |
| 18 | | researcher | ... | Workplace | workplace |
| 19 | | researcher | ... | Roles | project role |
| 20 | | researcher | ... | Workload | workload |
| 21 | | researcher | ... | Workload set | workload set |
| 22 | | risk | ... | Risk type | risk type |
| 23 | | risk | ... | Cost of risk (optional) | cost |
| 24 | | workload | ... | Project | project |
| 25 | | workload | ... | Project phase | project phase |
| 26 | | workload | ... | Researcher | researcher |
| 27 | | workload set | ... | Phase 1...7 | workload |
| 28 | Project Design Level, Resources Project Design | author | ... | Address | address |
| 29 | | author | ... | Researcher (optionally) | researcher |
| 30 | | cost | ... | Cost type | cost type |
| 31 | | experience | ... | Authors | author |
| 32 | | experience | ... | Type of experience | experience type |
| 33 | | resource | ... | Cost of resource (optional) | cost |
| 34 | | resource | ... | Resource type | resource type |
| 35 | | service | ... | Service cost | cost |
| 36 | | service | ... | Providing service | service group |
| 36 | | service | ... | Service group | service group |
| 37 | | technology | ... | Technology owner | workplace |
| 38 | | technology | ... | Technology authors | author |
| 39 | | technology | ... | Experience list | experience |
| 40 | | technology | ... | Resources | resource |
| 41 | | technology | ... | Services | service |
| 42 | | technology | ... | Category | technology category |
| 43 | | technology | ... | Requirement satisfied | requirement |
| 44 | Project Implementation Level, Relationships | requirement | ... | Author | researcher |
| 45 | | requirement | ... | 1.02 Requirement source | experience |
| 46 | | contain | ... | Type of relationship | relationship |
| 47 | | copy | ... | Type of relationship | relationship |
| 48 | | derive | ... | Type of relationship | relationship |
| 49 | | satisfy | ... | Type of relationship | relationship |
| 50 | | trace | ... | Type of relationship | relationship |
| 51 | | verify | ... | Type of relationship | relationship |
| 52 | Project Operational Level | project activity | ... | Task | project task |
| 53 | | project activity | ... | Researcher | researcher |
| 54 | | project activity | ... | Type of activity | project activity type |
| 55 | | task activity | ... | Project activity | project activity |
| 56 | | task activity | ... | Type of task activity | task activity type |

**Figure 17.** *ResearchML* abstraction layers, objects, relationships, target roles.

### 6.2.2. Project Design Level, Functional Project Design

Detailed information about the organization of project is found here (stereotypes: goal, goal type, project phase, project implementation rate, project rate type, project rate value, project role, project task, researcher, risk, risk type, workload, workload set). These allow to define the project's timetable,

team members, the workload of specific team members in this specific project as well as the other ones (that is, if the presented approach is applied to manage the portfolio of all projects).

### 6.2.3. Project Design Level, Resources Project Design

Detailed information about the resources used in the project (or projects); contains the following stereotypes: *author*, *cost*, *cost type*, *experience*, *experience type*, *resource*, *resource type*, *service*, *service group*, *technology*, *technology category*. In the adopted approach the key characteristics of the team is the set of technologies that it is acquainted with (*technology*) and as such can effectively apply them during the execution of the project. One such example may be the ability to conduct simulations of cyber-physical systems. In this case, the technology category (*technology category*) might be "computer simulation." The *experience* objects serve as a proof of expertise obtained in that area, while the *experience type* objects serve to categorize them (e.g., scientific articles, project reports, internet publications). Each such documented use of technology has its author or authors (*author*), thus it can be tracked to specific team members. Using a technology will often require the presence of resources (*resource*), often of different types (*resource type*), for example, software, hardware, chemical agents.

### 6.2.4. Project Implementation Level, Relationships

At this level, the *MagicGrid* process can be utilized; alternatively, *SysML* language or any domain-specific solution can be applied. Two stereotypes are utilized in *researchML* for model-based requirement management: *requirement* and *relationship.* The former is the general requirement object, the latter defines the relationships between requirements and architecture objects. Also, relationships extended from *SysML* v1.4 are implemented in the metamodel: *contain*, *copy*, *derive*, *satisfy*, *trace*, *verify*. They have the same meaning compared to *SysML* relationships with an extension of the *relationship* property.

*Project Implementation Level* may be realized with the use of *MagicGrid*, *SysML* or any other DSL. Preferably, it should be implemented with the 8-step workflow presented in Section 5.

### 6.2.5. Project Operational Level

In this abstraction level, the summary of activities undertaken during the project's execution is contained (stereotypes: *project activity*, *project activity type*, *task activity*, *task activity type*). With the developed report generators, the documentation management process is more effective. Team members devote less time for that task, thus the developed metamodel increases the team's overall productivity.

*ResearchML* is under intensive development. Another important area of abstraction that was added is *Project Implementation Level* which contains the requirement model. Due to the modularity of modelling languages, the *SysML* or any DSL could be used as a potential solution in this specific area. Also, a set of generators and data visualization algorithms are under development, including 2D/3D visual reports, similar to the ones used in Reference [89].

Project design template created with the use of *researchML* is shown in Figure 18.

### 6.3. Reporting, Specifications

Creation of modelling languages for multi-criterion and multi-stage projects is usually aided by advanced tools for data visualization and progress reporting. In this subsection, two important issues are addressed: generation of project's documentation and information exchange between models and their environment.

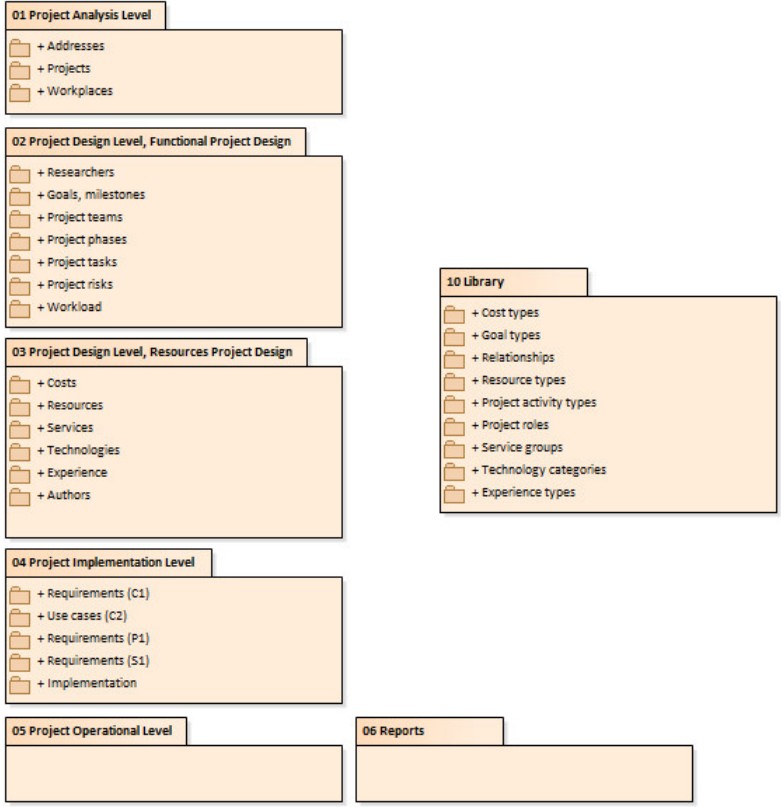

**Figure 18.** *ResearchML* project design template.

### 6.3.1. Documentation Generation

Generation of documentation follows the schema presented in Figure 19. The generators (including templates and format of generated files and documents) are an integral part of metamodel (M2). After the modelling language is defined, models and documentation models on the M1 level and their generation (M0), contribute to the robustness of project process against requirements changes.

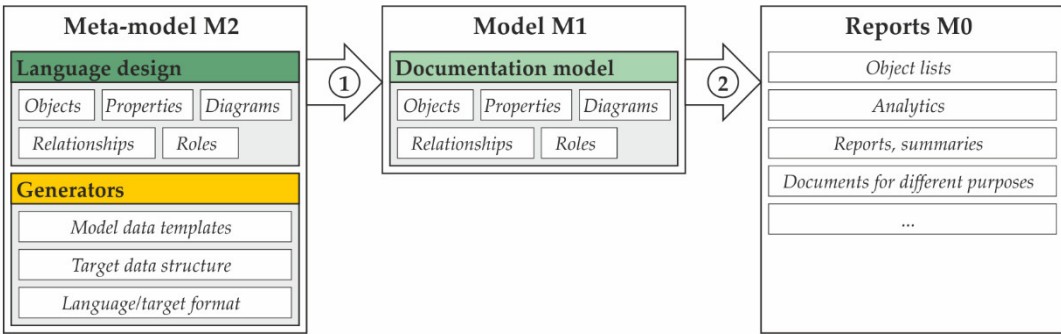

**Figure 19.** Reporting workflow.

Changes in documentation at the M0 level require manual updating of data that is contained within the model. Changes at the M1 level are visible in all places and perspectives of the modelled projects. It is important to delete the M0 documents when they are no longer needed—it is possible that such documents will no longer be consistent with data at the M1 level.

Generation of documentation with the approach presented in Figure 19 is the basis for automation of requirements analysis in control systems projects and products designed to meet the functional safety requirements [8,9,70,73,111,112].

### 6.3.2. Two-way Data Exchange with Model's Environment

Even the most advanced and automated procedure of documentation generation is still is a one-way process. Projects realized in teams often require a two-way data exchange between the model and external software solutions (e.g., other models created with the same language, simulation software, CAD software, control system design software). Whenever a necessity for actualization of large number of parameters exists, then a data exchange between the model and its environment is worth considering. The workflow of such integration is presented in Figure 20.

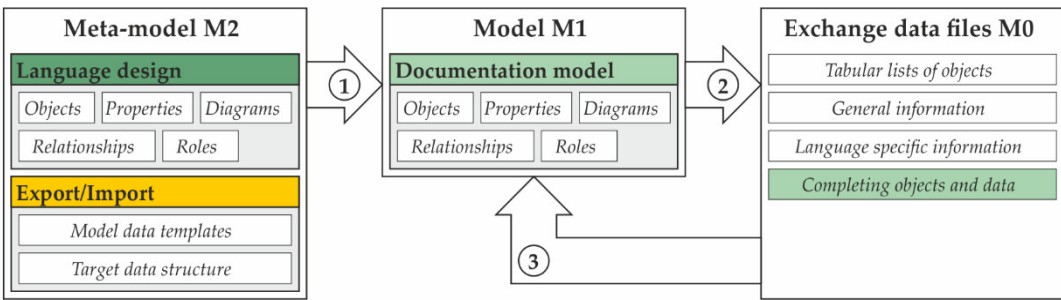

**Figure 20.** Specification workflow.

Metamodel (objects, properties, M2) should be designed in a way that allows straightforward import and export of large amounts of data by the means of M0-level files. Only then, the usage of modelling tool (M1) is effective.

The data exchange specification was defined at M2 level and not in generators. The developed modelling language (step 1 in Figure 20) defines data export (step 2 in Figure 20) from model (M1) to data files (M0) and allows information exchange between IT systems and other software solutions.

Data files (M0) may be actualized or updated with new objects; importing the data from an external source (step 3 in Figure 20) causes actualization of all objects present in the model. This could also lead to creation of new objects. It is of upmost importance to be careful during the data import. A mass import of new objects or actualization of the already existing ones will most likely lead to major problems, (e.g., loss of information in the model).

## 7. Conclusions

To summarize, *Model-Based Design* is very useful for designing mechatronic and cyber-physical systems. It expands upon the *Model-Driven Engineering* paradigm and allows for effective solving of problems in prototyping, development, standardization and management of projects—issues that often haunt the creation of such systems.

*Model-Based Design* supports all areas of systems' development allowing for employing advanced data analytics. A novelty in this approach is the analysis of relations directly between the objects and not only their parameters. This approach in a much greater scale leads to fulfilment of business goals that are set for such projects.

However, implementing the *Model-Based Design* in team environment is not an easy task. Prior experience in document-based approach leads to reluctance when exploring the possibility of shifting to model-based development, even though the latter offers an increase in overall productivity. On the way to full implementation of the *Model-Based Design* paradigm, the human factor may be the hardest to overcome. The cultural, nationality and education backgrounds, as well as experience of the engineers are major issues whenever a practical implementation takes place. All of these problems were observed during the realization of the iLOAD project.

Another issue is a high initial cost, usually in three areas: (1) software and hardware, (2) training, (3) external consultations. That last problem stems from the fact that it is highly unlikely that a major shift (i.e. adoption of new methodology) is initiated from within the company—it often lacks the

qualified personnel. It is also the reason why projects involving the academia and industry partners are so valuable.

Also, the first problem may be somewhat averted by adopting the MBD approach in a systematic fashion. Beginning from MBD-1 (Figure 1) up to MBD-3, then moving to MBD-4 and MBD-6. Finally, reaching the last layer concerned with the requirements management and moving forward on it (MBD-7 to MBD-9). The process, however, should not be rushed—time is needed for the methodology to take roots and for the processes to be understood by all team members. Only then, any investments in software, hardware, training and external consulting can be considered. The initial reluctance must also be combatted by appointing project leaders that are willing to put through the new approach in a controlled and organized manner.

As a result, the author expects that the methodology presented in this paper will be increasingly popular in management of cyber-physical and mechatronic projects—especially in mixed teams where representatives from academia and industry are present. The problem of designing functional system safety or more generally, multi-criterion constraints that stem from multiple sources are the new challenges for MBD. These problems are essential when considering the rise of Industry 4.0, development of autonomous vehicles and cooperative robotics.

In this work, a relatively novel approach for scientific research was also presented – *Model-Based Management*. A relational approach allows for tackling the problems that were previously out of reach, for example, similarity analysis of chosen safety functionalism. The strengths of this approach can be attributed to an effective analysis of large data sets.

This paper also confirms the scalability of the approach for more abstract problems, that is, from operational to strategic planning. The implementation of *researchML* language uses *SysML* language or other *Domain-Specific Languages* that are compatible with the 8-steps workflow presented in the discussion (Section 5).

When applying *researchML* to the strategic management of projects, the following workflows can be considered:

1. Top-down—from more abstract layers/levels to precise definitions of properties and objects;
2. Bottom-up—from precise definitions of properties and objects to more abstract levels including project architecture;
3. Free—new objects are created when needed;

Finally, the *researchML* implementation fully supports agile project management. It is a big advantage of the proposed modelling language and is caused by the domain-specific approach to its design.

Concepts, workflows, methods and language *researchML* presented in the paper are applied in current projects: "Development of devices for medical rehabilitation supported by telemedicine (ARM)" (EU co-financing under grant agreement no. RPZP.01.01.00-32-0030/17), "RapidSteel—development of an integrated chain of tools to optimize and automate the design process and robotize the production of heavy-duty components (RapidSteel)" (EU co-financing under grant agreement no. POIR.01.01.01-00-0441/18), "ZUT 2.0—Modern Integrated University" (EU co-financing under grant agreement no. POWR.03.05.00-00-Z205/17) and "Development of a model concept for the categorization and parameterization of elements of the digital innovation hub network (DIH) and a feasibility study" (grant financed by the Ministry of Entrepreneurship and Technology, Poland). In forthcoming works author of presented here paper is planning to present newly proposed metamodel RAMIF (*Reference Architecture Metamodel for Industry of the Future*). It will be application of RAMI 4.0 metamodel for discussion on Industry 4.0 complexity issues in Poland.

**Funding:** Presented here research received no external funding. It is inspired, among the others by the works initiated in project "iLoad – Partnership for developing energy efficient intelligent load handling system" Marie Curie 7PR, FP7-PEOPLE-2012-IAPP (Industry-Academia Partnerships and Pathways) grant number 324496 and "Family of high-performance, universal 5-axis machining centers type X-5" Innotech In-Tech, research and development grant number 158356.

**Conflicts of Interest:** The author declares no conflict of interest.

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
