# Peer review of "Metamodelling for Design of Mechatronic and Cyber-Physical Systems"

_applsci, doi:10.3390/app9030376_

Round 1

Reviewer 1 Report

This paper titled “Meta-modeling of complex mechatronic systems” deals with the most relevant aspects related to the modelling phases of a mechatronics design of complex systems. In particular, this paper is divided into three parts: in the first one two design processes are presented and described, while in the second fundamental aspects of meta-modelling as a process of creating domain-specific languages are presented. Finally, an example of modelling language is introduced.

The paper is reasonably clear but in my opinion not well structured. In fact, the third part is presented in an appendix, while the first and the second are developed in two separate sections without a general overview of the proposed approach. My suggestion is add also a section titled “Conclusions” in order to resume and conclude the work.

The result of this is that the paper seems to be a superficial description of a design procedure of a complex system and not a scientific paper useful for actual design tasks.

Moreover, in the paper titled

“Domain specific language for structural modeling of logically controlled discrete-event mechatronic systems”, (2017) 22nd International Conference on Methods and Models in Automation and Robotics, MMAR 2017, art. no. 8046788, pp. 1-6. DOI: 10.1109/MMAR.2017.8046788

recently published by Pietrusewicz, K., Scopchanov, M., the authors focus on the language used to describe a model of a controlled mechatronics system. This research is strictly related with the proposed one and I wonder why the author doesn’t present it and analyse in relation to the new version.

In the “Domain specific language for structural modeling of logically controlled discrete-event mechatronic systems” paper the authors show the use of “their language” by an example. It is my considered opinion that also in this new work the author should use an example to better explain the elements of novelty of its approach.

Furthermore, it would be useful to compare the proposed procedure with alternative approaches available in literature such as:

1.       Moulianitis, V.C., Zachiotis, G.-A.D., Aspragathos, N.A., A new index based on mechatronics abilities for the conceptual design evaluation, (2018) Mechatronics, 49, pp. 67-76, DOI: 10.1016/j.mechatronics.2017.11.011

2.       Sadlauer, A., Hehenberger, P.”Using design languages in model-based mechatronic system design processes” (2017) International Journal of Agile Systems and Management, 10 (1), pp. 73-91. DOI: 10.1504/IJASM.2017.082940

3.       Zheng, C., Hehenberger, P., Le Duigou, J., Bricogne, M., Eynard, B., Multidisciplinary design methodology for mechatronic systems based on interface model, (2017) Research in Engineering Design, 28 (3), pp. 333-356. DOI: 10.1007/s00163-016-0243-2

4.       Hirz, M., An approach supporting integrated modeling and design of complex mechatronics products by the example of automotive applications, (2018) WMSCI 2018 - 22nd World Multi-Conference on Systemics, Cybernetics and Informatics, Proceedings, 3, pp. 161-166.

5.       Escobar, L., Carvajal, N., Naranjo, J., Ibarra, A., Villacis, C., Zambrano, M., Galarraga, F., “Design and implementation of complex systems using Mechatronics and Cyber-Physical Systems approaches” (2017) 2017 IEEE International Conference on Mechatronics and Automation, ICMA 2017, art. no. 8015804, pp. 147-154, DOI: 10.1109/ICMA.2017.8015804

6.       Guerineau, B., Bricogne, M., Durupt, A., Rivest, L., Mechatronics vs. cyber physical systems: Towards a conceptual framework for a suitable design methodology, (2016) 2016 11th France-Japan and 9th Europe-Asia Congress on Mechatronics, MECATRONICS 2016 / 17th International Conference on Research and Education in Mechatronics, REM 2016, art. no. 7547161, pp. 314-320, DOI: 10.1109/MECATRONICS.2016.7547161

Finally, the analysis of mechatronics designs available in scientific literature could be useful to better understand the improvement obtainable with the method proposed by the author. In other terms how the use of the proposed method would have improved works such as:

1.       Vulliez, P., Gazeau, J.P., Laguillaumie, P., Mnyusiwalla, H., Seguin, P, Focus on the mechatronics design of a new dexterous robotic hand for inside hand manipulation, (2018) Robotica, 36 (8), pp. 1206-1224, DOI: 10.1017/S0263574718000346

2.       Handono, K., Sumarno, E., Haryanto, D., Kiswanta, Setiadipura, T., Indrakoesoema, K., Rokhmadi, Mechatronic design and analysis of reaktor daya experimental components, (2018) International Journal of Mechanical Engineering and Technology, 9 (9), pp. 405-414.

3.       Giberti, H., Sbaglia, L., Silvestri, M. Mechatronic design for an extrusion-based additive manufacturing machine, (2017) Machines, 5 (4), art. no. 29, DOI: 10.3390/machines5040029

For these reasons and in the present form, it is my considered op

Author Response

Dear Reviewer,

I am very grateful for the insightful remarks. The article has been corrected as per your observations, and hopefully, is a much more valuable research study now. My response is given in general for the major points included in the review. I hope you will be satisfied with the proposed changes.

Main language update is (due to additional discussion with my colleagues at the University and literature review) that in whole paper the following naming convention is used: meta-metamodel (M3), metamodel (M2), model (M1). Previously I used meta-model (M2) naming convention.

Also the title is updated to Metamodeling for design of mechatronic and cyber-physical systems

Best regards,

Krzysztof Pietrusewicz

Pietrusewicz Krzysztof, PhD, DSc, Associate Professor

Rector's representative for Industry 4.0

West Pomeranian University of Technology, Szczecin

Faculty of Electrical Engineering

Department of Control Engineering and Robotics

Sikorskiego 37, 70 – 313 Szczecin, Poland

Mobile: +48 663 398 396

Hangouts, Keep: krzysztof.pietrusewicz.kp@gmail.com

Skype: krzysztof.pietrusewicz

Linkedin: https://www.linkedin.com/in/krzysztof-pietrusewicz-a6666278

Reviewer 2 Report

Generally a very interesting topic – and a solid basis for research.

However, some major weaknesses prohibit a publication in this form:

The use of articles and the language needs to be checked by a native speaker.

Many current trends of model based systems engineering are missing such as
the use of graph-based languages (see e.g.

Groß, J.; Rudolph, S.: Generating simulation models from UML – a FireSat example. In: Proceedings of the 2012 Symposium on Theory of Modeling and Simulation – DEVS Integrative M&S Symposium. San Diego: Society for Computer Simulation International, 2012

Holder, K.; Zech, A.; Ramsaier, M.; Stetter, R.; Niedermeier, H.-P.; Rudolph, S.; Till, M.: Model-Based Requirements Management in Gear Systems Design Based On Graph-Based Design Languages: In. Applied Sciences. 2017, 7, 1112; doi:10.3390.

Patrice Micouin, Model Based Systems Engineering: Fundamentals and Methods, 2014

Estefan, Jeff A. "Survey of model-based systems engineering (MBSE) methodologies." Incose MBSE Focus Group 25 (2007): 8 )

The review of MBSE has to be much more far-reaching and concise

The authors are able to mention three applications:

motion control algorithm of 5-axis feed drive "X-5" processing center owned by the AVIA 9 company,

loading crane control system designed within the EU 7FP iLoad project
and

control 10 system of forestry crane - HIAB HiVision

the authors need to present at least one example in detail in the paper in order to validate their approach – the author should really presents results such as the functional and risk analysis and the system context representation – at least an excerpt of this.

The authors need to report the experience of the involved engineers - did the meta-modeling really support their processes? How did it support it? Were some current weaknesses identified?

All three examples focus on control – many other components of complex mechatronic systems are not addressed. Are the scientific claims of the authors only related to control? Or are the authors able to illustrate an application to other components in the interesting featured applications?

Author Response

(The authors gave the same response as above.)

Round 2

Reviewer 1 Report

The paper has been change following the main recommendations and it is my considered opinion that now in its present form it is publishable.